

**1** **Anthropogenic CO$_2$, air-sea CO$_2$ fluxes and acidification in the Southern**
**2** **Ocean: results from a time-series analysis at station OISO-KERFIX (51°S-**
**3** **68°E).**

Nicolas Metzl[1], Claire Lo Monaco[1], Coraline Leseurre[1,2], Céline Ridame[1], Gilles Reverdin[1],
Thi Tuyet Trang Chau[3], Frédéric Chevallier[3], Marion Gehlen[3]
[1] Laboratoire LOCEAN/IPSL, Sorbonne Université-CNRS-IRD-MNHN, Paris, 75005, France
[2] Flanders Marine Institute (VLIZ), 8400 Ostend, Belgium
[3] Laboratoire LSCE/IPSL, CEA-CNRS-UVSQ, Université Paris-Saclay Gif-sur-Yvette, 91191, France
*Correspondence to*: Nicolas Metzl (nicolas.metzl@locean.ipsl.fr)
**Abstract:** The temporal variation of the carbonate system, air-sea CO$_2$ fluxes and pH is analyzed in the Southern
Indian Ocean, south of the Polar Front, based on in-situ data obtained from 1985 to 2021 at a fixed station
(50°40'S-68°25'E) and results from a neural network model that reconstructs the fugacity of CO$_2$ (fCO$_2$) and
fluxes at monthly scale. Anthropogenic CO$_2$ (C$_{ant}$) was estimated in the water column and detected down to the
bottom (1600m) in 1985 resulting in an aragonite saturation horizon at 600m that migrated up to 400m in 2021
due to the accumulation of C$_{ant}$. In subsurface, the trend of C$_{ant}$ is estimated at +0.53 (±0.01) µmol.kg$^{-1}$.yr$^{-1}$ with a
detectable increase in recent years. At the surface during austral winter the oceanic fCO$_2$ increased at a rate close
or slightly lower than in the atmosphere. To the contrary, in summer, we observed contrasting fCO$_2$ and
dissolved inorganic carbon (C$_T$) trends depending on the decade and emphasizing the role of biological drivers
on air-sea CO$_2$ fluxes and pH inter-annual variability. The region moved from an annual source of 0.8 molC.m$^{-}$
$^2$.yr$^{-1}$ in 1985 to a sink of -0.5 molC.m$^{-2}$.yr$^{-1}$ in 2020. In 1985-2020, the annual pH trend in surface of -0.0165 (±
0.0040).decade$^{-1}$ was mainly controlled by anthropogenic CO$_2$ but the trend was modulated by natural processes.
Using historical data from November 1962 we estimated the long-term trend for fCO$_2$, C$_T$ and pH confirming
that the progressive acidification was driven by atmospheric CO$_2$ increase. In 59 years this leads to a diminution
of 11% for both aragonite and calcite saturation state. As atmospheric CO$_2$ will desperately continue rising in the
future, the pH and carbonate saturation state will decrease at a faster rate than observed in recent years. A
projection of future C$_T$ concentrations for a high emission scenario (SSP5-8.5) indicates that the surface pH in
2100 would decrease to 7.32 in winter. This is up to -0.86 lower than pre-industrial pH and -0.71 lower than pH
observed in 2020. The aragonite under-saturation in surface waters would be reached as soon as 2050 (scenario
SSP5-8.5) and 20 years later for a stabilization scenario (SSP2-4.5) with potential impacts on phytoplankton
species and higher trophic levels in the rich ecosystems of the Kerguelen Island area.
Keywords: Ocean Carbonate System, Ocean acidification, anthropogenic CO$_2$, air-sea CO$_2$ fluxes, Southern
Ocean, Time-series station





**1 Introduction**

The ocean plays an important role in mitigating climate change by taking up since decades a large part

of the excess of heat (Cheng et al., 2020; Fox-Kemper et al., 2021) and of $CO_2$ released by human activities
(Sabine et al., 2004; Gruber et al., 2019a; Canadell et al., 2021). Since 1750, the global ocean has captured 185
($\pm$ 35) PgC (Petagramm of Carbon) from a total of 700 ($\pm$ 75) PgC of anthropogenic carbon emissions from
fossils fuels and land-used changes (Friedlingstein et al., 2022). From year to year, the ocean anthropogenic $CO_2$
sink increased progressively from 1.1 ($\pm$ 0.4) $PgC.yr^{-1}$ in the 1960s to 2.3 ($\pm$ 0.4) $PgC.yr^{-1}$ in the 2000s. Over the
decade 2012-2021, the partitioning of the anthropogenic $CO_2$ sinks was roughly equal between the ocean (2.9 $\pm$
0.4 $PgC.yr^{-1}$) and the land (3.1 $\pm$ 0.6 $PgC.yr^{-1}$) (Friedlingstein et al., 2022).

Ocean observations indicate that since the 1990s the Southern Ocean (SO) south of 45°S has been

accumulating each year about 0.5 $PgC.yr^{-1}$ (e.g. Takahashi et al., 2009; Lenton et al., 2013; Rödenbeck et al.,
2013; Long et al., 2021; Fay et al, 2023; Gray, 2024). Results based on BGC-Argo floats (Southern Ocean
Carbon and Climate Observations and Modeling project, SOCCOM) suggest that the $CO_2$ sink in the SO might
be much lower (0.16 $PgC.yr^{-1}$ south of 44°S for the period 2015-2017, Gray et al. 2018; Bushinsky et al., 2019)
but there is an ongoing debate (Long et al., 2021; Sutton et al., 2021; Gray, 2024). It is also well established that
the $CO_2$ sink in the SO undergoes substantial decadal variability first documented for the 1990s (Le Quéré et al.,
2007; Metzl, 2009; Lenton et al., 2013) and subsequently identified for the period 1982-2018 (Landschützer et
al., 2015; Keppler and Landschützer, 2019; Mackay et al., 2022; Hauck et al., 2023). A recent extension of the
period to 1957-2020 suggests that the inter-annual to decadal variability of the SO $CO_2$ sink was most
pronounced after the 1980s (Rödenbeck et al., 2022; Bennington et al., 2022). Whatever the variability of the SO
$CO_2$ sink since the 1960s, the ocean continuously absorbs atmospheric $CO_2$ and the distribution of anthropogenic
$CO_2$ ($C_{ant}$) in the SO is now relatively well documented (e.g. Pardo et al., 2014; Gruber et al., 2019a) thanks to
the GLODAP data synthesis effort for the global ocean (Global Ocean Data Analysis Project, Olsen et al., 2016,
2019, 2020). The SO takes up about 40% of the total anthropogenic carbon that enters the ocean (Khatiwala et
al., 2013; Gruber et al., 2019a).

The anthropogenic $CO_2$ uptake in the ocean results is lowering carbonate ion concentrations and pH, a

chemical process termed "ocean acidification" (OA) (Caldeira and Wickett 2003; Doney et al., 2009). This
decreases the saturation state with respect to carbonate minerals (aragonite, $\Omega ar$ and calcite, $\Omega ca$), a process most
pronounced in the cold and naturally at a low saturation state waters in high latitudes (Orr et al., 2005; Takahashi
et al., 2014; Jiang et al., 2015). The first estimate of $C_{ant}$ distribution in the global ocean (for a nominal year
1994, Sabine et al., 2004) shows that $C_{ant}$ uptake led to an upward migration of the $\Omega ar$ and $\Omega ca$ saturation
horizon in all ocean basins (Feely et al., 2004). This change is particularly pronounced south of the Polar Front
(PF) in the SO linked to both $C_{ant}$ uptake and the upwelling of dissolved inorganic carbon ($C_T$) $C_T$-rich deep
waters (e.g. Hauck et al., 2010; Pardo et al., 2017). It has been suggested, through numerical studies, that
depending on future $CO_2$ emission levels, surface waters could reach under-saturation state for aragonite by
2030-2050 in the SO (Orr et al., 2005; Gangstø et al., 2008; McNeil and Matear, 2008; Negrete-Garcia et al.,
2019). Such a change would have multiple and detrimental impacts on marine ecosystems (Fabry et al., 2008;
Doney et al., 2012; Bopp et al., 2013), in particular calcifying marine organisms especially aragonite producers
such as pteropods (Hunt et al., 2008: Gardner et al., 2023), but also calcite producing planktonic foraminifera
(Moy et al., 2009), coccolithophorids (Beaufort et al., 2011), and non-calcifying species such as the abundant SO



diatoms (e.g. Benoiston et al., 2017; Petrou et al., 2019; Weir et al., 2020; Duncan et al., 2022) and krill
(Kawaguchi et al., 2013).
Hindcast simulations with Global Ocean Biogeochemical Models (GOBM), as well as projections with
Earth System Models (ESM) have been used to evaluate the ocean carbon cycle over the past decades and future
changes in $C_{ant}$ storage, ocean acidification or impacts of global change on marine ecosystems. However, current
model-based estimates of the contemporary SO $CO_2$ sink are subject to relatively large uncertainties (e.g. Long
et al., 2013; Hauck et al., 2020; Gooya et al., 2023; Hauck et al., 2023; Mayot et al., 2023; DeVries et al, 2023).
Difference between GOBM models can reach 0.7 $PgC.yr^{-1}$ in the SO (Hauck et al., 2020), which is roughly
equivalent to the mean climatological flux of 0.5 $PgC.yr^{-1}$ (McNeil et al., 2007; Takahashi et al., 2009; Lenton et
al., 2013). In the high latitudes of the SO (> 50°S) for the 2010s, ESM from the Coupled Model Intercomparison
Project Phase 6 (CMIP6) simulated either a large sink or a modest source of $CO_2$ (McKinley et al, 2023). This is
mainly due to incorrect or missing physical and/or biological processes in the models (e.g. Pilcher et al., 2015;
Kessler and Tjiputra, 2016; Mongwe et al., 2018; Lerner et al., 2021) leading to biases in the seasonality of
temperature, $C_T$, partial pressure of $CO_2$ ($pCO_2$), air-sea $CO_2$ fluxes, pH or $\Omega$ (e.g. McNeil and Sasse 2016;
Rodgers et al., 2023; Rustogi et al., 2023; Joos et al., 2023). Such model imperfections should be resolved to
have future projections of $CO_2$ uptake, OA, productivity and the responses of the marine ecosystems, gain in
reliability (Frölicher et al., 2015; Hauck et al., 2015; Sasse et al., 2015; Kessler and Tjiputra, 2016; McNeil and
Sasse 2016; Kwiatkowski and Orr, 2018; Negrete-Garcia et al., 2019; Burger et al., 2020; Terharr et al., 2021;
Krumhardt et al., 2022; Jiang et al., 2023; Mongwe et al., 2023). In this context, as often concluded in modeling
studies (e.g. Kessler and Tjiputra, 2016; Gooya et al., 2023; Wright et al., 2023; Hauck et al., 2023; Mayot et al.,
2023; Rodgers et al., 2023), long-term biogeochemical observations are particularly valuable to quantify and
understand recent past and current changes, and ultimately evaluate model simulations.
Although the SO south of the Polar Front remains much less observed than other oceanic regions,
several observations based studies have allowed to estimate the decrease in pH in the surface waters in response
to the increase in oceanic $CO_2$ fugacity, $fCO_2$ (Mirodikwa et al., 2012; Takahashi et al., 2014; Lauvset et al.,
2015; Munro et al., 2015; Xue et al., 2018; Iida et al., 2021; Leseurre et al., 2022; Brandon et al., 2022). Results
showed a large range of the pH trends from -0.008.$decade^{-1}$ to -0.035.$decade^{-1}$ depending on the period and
regions of interest. Most of these analyses were based on summer observations (Table 1) and some studies
highlighted contrasting pH trends on a 5-10 year time probably linked to large scale climate variability such as
the Southern Annular Mode (SAM) (e.g. Xue et al., 2018). Given such variability, it is important to continue
monitoring $fCO_2$ and pH trend and, if possible, at different seasons as future change in $CO_2$ uptake and potential
tipping points of that carbonate saturation state also depends on seasonality (Sasse et al., 2015). The above
observational studies were dedicated to pH changes in surface waters. In contrast to Northern high latitudes (e.g.
Olafsson et al., 2009, 2010; Franco et al., 2021; Skjelvan et al., 2022) few studies in the SO attempted to
evaluate decadal changes of carbonate system properties and acidification in the water column based on time-
series stations. These changes in the SO water column were investigated from data collected during cruises
generally 3 to 15 years apart (Hauck et al., 2010; Van Heuven et al., 2011; Pardo et al., 2017; Tanhua et al.,
2017; Carter et al., 2019).



Table 1: Trends of oceanic $fCO_2$ ($\mu atm.yr^{-1}$) and pH ($decade^{-1}$) in the Southern Ocean south of the Polar Front based on observations and from this study. IO: Indian Ocean sector. PO: Pacific Ocean sector. AO: Atlantic Ocean sector. SO SPSS: Southern Ocean SubPolar Seasonally Stratified biome (around 50-60°S). PZ: Polar Zone. NR: Not Reported. Standard-deviations when available are in bracket.

| Period | Season | Zone | Trend $fCO_2$ $\mu atm.yr^{-1}$ | Trend pH $decade^{-1}$ | Reference |
|---|---|---|---|---|---|
| 1991-2000 | Summer | IO PZ 55-60°S | 2.93 | -0.035 | Xue et al (2018) |
| 2001-2011 | Summer | IO PZ 55-60°S | 1.41 | -0.016 | Xue et al (2018) |
| 2005-2019 | Summer | IO PZ 54-64°S | NR | -0.026(0.003) | Brandon et al (2022) |
| 1998-2019 | Summer | IO 50°S-68°E | 1.9 (0.3) | -0.019 (0.004) | Leseurre et al (2022) |
| 1998-2019 | Summer | IO 55°S-63°E | 2.1 (0.3) | -0.022 (0.003) | Leseurre et al (2022) |
| 1998-2007 | Summer | IO 55°S-63°E | 5.3 (0.4) | -0.050 (0.016) | Leseurre et al (2022) |
| 2006-2019 | Summer | IO 55°S-63°E | 0.3 (0.2) | no trend | Leseurre et al (2022) |
| 1969-2003 | Summer | PO 55-62°S | 1.7 (0.2) | -0.020 (0.003) | Midorikawa (2012) |
| 2002-2012 | Annual | Drake North | 2.21 (0.55) | -0.023 (0.007) | Takahashi (2014) |
| 2002-2012 | Annual | Drake South | 1.50 (0.65) | -0.015 (0.008) | Takahashi (2014) |
| 2002-2015 | Summer | Drake North | 1.95 (0.55) | -0.021 (0.006) | Munro et al (2015) |
| 2002-2015 | Winter | Drake North | 1.92 (0.24) | -0.018 (0.003) | Munro et al (2015) |
| 2002-2015 | Summer | Drake South | 1.30 (0.85) | -0.017 (0.010) | Munro et al (2015) |
| 2002-2015 | Winter | Drake South | 0.67 (0.39) | -0.008 (0.004) | Munro et al (2015) |
| 2002-2015 | Annual | Drake North | 1.74 (0.15) | -0.019 (0.002) | Munro et al (2015) |
| 2002-2015 | Annual | Drake South | 1.16 (0.27) | -0.015 (0.003) | Munro et al (2015) |
| 1981-2011 | Annual | SO SPSS | 1.44 (0.10) | -0.020 (0.002) | Lauvset et al (2015) |
| 1991-2011 | Annual | SO SPSS | 1.46 (0.11) | -0.021 (0.002) | Lauvset et al (2015) |
| 1993-2018 | Annual | SO 44-75°S | NR | -0.0165 (0.0001) | Iida et al (2021) |
| 1962-2016 | November | IO 50°S-68°E | 1.31 (0.20) | -0.014 (0.002) | This study, Obs. |
| 1991-2021 | Summer | IO 50°S-68°E | 2.10 (0.22) | -0.022 (0.002) | This study, Obs. |
| 1991-2001 | Summer | IO 50°S-68°E | 0.76 (0.90) | -0.009 (0.010) | This study, Obs. |
| 2001-2010 | Summer | IO 50°S-68°E | 3.23 (1.07) | -0.035 (0.011) | This study, Obs. |
| 2010-2020 | Summer | IO 50°S-68°E | 0.84 (0.77) | -0.008 (0.008) | This study, Obs. |
| 1985-2020 | Summer | IO 50°S-68°E | 1.71 (0.08) | -0.018 (0.001) | This study, FFNN |
| 1991-2020 | Summer | IO 50°S-68°E | 1.85 (0.11) | -0.020 (0.001) | This study, FFNN |
| 1991-2001 | Summer | IO 50°S-68°E | 1.18 (0.26) | -0.013 (0.004) | This study, FFNN |
| 2001-2010 | Summer | IO 50°S-68°E | 2.87 (0.25) | -0.030 (0.003) | This study, FFNN |
| 2010-2020 | Summer | IO 50°S-68°E | 0.98 (0.40) | -0.010 (0.004) | This study, FFNN |
| 1991-2001 | Winter | IO 50°S-68°E | 0.98 (0.09) | -0.010 (0.001) | This study, FFNN |
| 2001-2010 | Winter | IO 50°S-68°E | 1.99 (0.10) | -0.021 (0.001) | This study, FFNN |
| 2010-2020 | Winter | IO 50°S-68°E | 2.21 (0.17) | -0.022 (0.002) | This study, FFNN |
| 1985-2020 | Annual | IO 50°S-68°E | 1.57 (0.03) | -0.0165(0.0004) | This study, FFNN |

The present study complements in time, seasons, and in the water column, the surface $fCO_2$ and pH trends investigated by Leseurre et al., (2022) in different regions of the Southern Indian Ocean for the period 1998-2019 during austral summer. South of the PF around 50°S, Leseurre et al. (2022) showed that in summer the surface $fCO_2$ increase and pH decrease over 20 years were mainly driven by the increase in anthropogenic $CO_2$ sequestration about +0.6 (± 0.2) $\mu mol.kg^{-1}.yr^{-1}$ and by a small warming of +0.03 (± 0.02) $°C.yr^{-1}$. In addition Leseurre et al. (2022) showed that in the recent decade, 2007-2019, the $fCO_2$ trend was low +0.3 (± 0.2) $\mu atm\ yr^{-1}$ compared to +5.3 (± 0.4) $\mu atm\ yr^{-1}$ in 1998-2007, highlighting the sensitivity of the $fCO_2$ and pH trends to the



selected time period (especially during summer). In particular, they observed relatively stable pH values over
2010-2019 (i.e. no decrease in pH) with no clear explanation on the origin of the slow-down of the $fCO_2$ and pH
trends in surface waters south of the PF in recent years. To complement the analysis by Leseurre et al. (2022)
based on summer observations in 1998-2019 this study focusses on one location regularly visited south of the
Polar Front (around 50°S-68°E south-west of Kerguelen Island, Figure 1). The analysis period is extended back
to 1985 and forward to 2021 to investigate the recent status of $fCO_2$ and pH. We also evaluate the trends for
different seasons using sparse spring/winter data. The combination of in situ observations and monthly estimates
from a neural network model over the period 1985-2020 (Chau et al., 2022) enables to assess potential changes
in seasonality of the surface ocean carbonates system (including $fCO_2$, $C_T$, pH, $\Omega$) as suggested in recent decades
or in future scenarios (Gallego et al., 2018; Landschützer et al., 2018; Kwiatkowski and Orr, 2018; Kwiatkowski
et al., 2020; Lerner et al., 2021; Fassbender et al., 2022; Yun et al., 2022; Rodgers et al., 2023; Joos et al., 2023).
The variability in surface waters will be related to changes in $C_{ant}$ concentrations observed in the water column
and will be complemented by an analysis of OA at depth between 1985 and 2021. Finally we will explore the
long-term variability of $fCO_2$ and pH since 1960s and potential future changes of the carbonate system at this
time series site.

**2 Data selection, methods and quality control**

**2.1 Study area and data selection**

This study focused on a High Nutrients Low Chlorophyll area (HNLC, Minas and Minas, 1992) of the

Indian sector of the Southern ocean (SO) in the Permanent Open Ocean Zone (POOZ) south of the Polar Front
(PF) and south-west of Kerguelen Islands (around 50°S-68°E, Figure 1). The Kerguelen Plateau is an extended
topographic feature that controls part of the Antarctic Circumpolar Current (ACC), generates eddies (Daniault
and Ménard, 1985) and the northward deflection of the PF near the Island (Pauthenet et al., 2018). The Plateau is
also a region of relatively high Chl-a (Moore and Abbott, 2000; Mongin et al., 2008) and strong $CO_2$ uptake
during austral spring-summer that contrasts with the weaker sink over the POOZ/HNLC (Metzl et al., 2006;
Jouandet et al., 2008, 2011; Lo Monaco et al., 2014; Leseurre et al., 2022). The POOZ/HNLC region west
(upstream) of the Kerguelen Plateau is characterized by rather stable water mass properties (temperature,
salinity, oxygen or nutrients) over time and low eddy activity compared to the Plateau (Daniault and Ménard,
1985; Chapman et al., 2015; Dove et al., 2022). In this region, located in the deep Enderby Basin, the flow is not
constrained by topography and there is no local upwelling that would import $C_T$-rich water to surface layers as
observed on the eastern side of the Kerguelen Plateau (Brady et al., 2021).

The Indian austral sector is also recognized to host the strongest winds in the SO leading to year-round

high gas transfer coefficients (Wanninkhof and Trinanes 2017). As a result, and in contrast to the Atlantic sectors
of the SO, the Indian region south of 45°S was a periodic $CO_2$ source, especially in the 1960s to the 1980s
(Rödenbeck et al., 2022; Bennington et al., 2022; Prend et al., 2022; Gray, 2024). In the POOZ-HNLC region,
high winter wind speed (monthly average up to 16 m.s$^{-1}$) and associated heat loss drive deep mixing. Deep
winter mixing entrains subsurface properties to the surface layer, increases $C_T$ concentrations leading to
wintertime outgassing of $CO_2$ (Metzl et al., 2006). This combination of characteristics makes the region an ideal
test-bed for 1-D modeling studies investigating the temporal dynamics and drivers of biogeochemical processes



including nutrients, iron, phytoplankton and carbon (Pondaven et al., 1998, 2000; Louanchi et al., 1999, 2001; Jabaud-Jan et al., 2004; Metzl et al., 2006; Mongin et al., 2006, 2007; Kane et al., 2011; Pasquer et al., 2015; Demuynck et al., 2020).

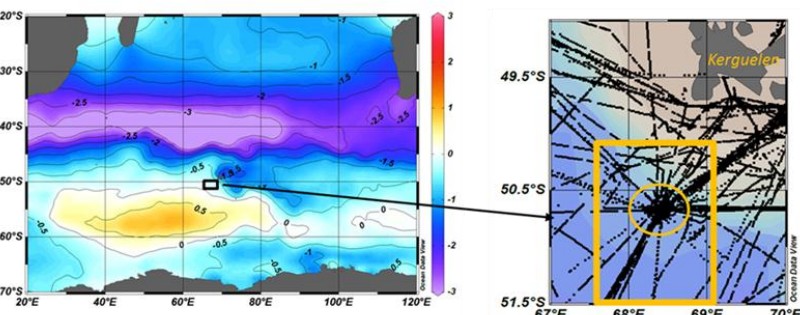

Figure 1: Left: Annual air-sea $CO_2$ flux (molC.m$^{-2}$.yr$^{-1}$) in the South Indian Ocean for year 2020 from the FFNN model (negative flux for ocean sink, positive flux for ocean source). The black box identified the location of the study south-west of Kerguelen. Right: Track of cruises with underway $fCO_2$ data South-West of Kerguelen Island. The station at 50°40'S-68°25'E occupied in 1985, 1992-1993 and 1998-2021 is indicated by a yellow circle. The yellow square is the region selected to calculate the mean values from the underway surface observations and from the FFNN model. Figures produced with ODV (Schlitzer, 2018).

Here we used surface and water-column observations around location 50°40'S-68°25'E (Figure 1, Table S1), historically called KERFIX station (KERguelen FIXed station) sampled in 1990-1995 in the framework of the WOCE/JGOFS programs (Jeandel et al., 1998). The station was first occupied in March 1985 during the INDIGO-1 cruise (Indian Ocean Geochemistry, Poisson, 1985; Poisson et al., 1988) and since 1998 it is regularly visited during the OISO cruises (Océan Indien Service d'Observations, Metzl and Lo Monaco, 1998, https://doi.org/10.18142/228). The regular occupation from 1985 to 2021 makes it the longest time-series station in the Southern Ocean POOZ/HNLC area allowing investigating the inter-annual to decadal trends of carbonate properties in surface waters and across the water-column (0-1600m). Despite the occasional variability in surface waters properties (e.g. lower surface Salinity in 2011-2013) we consider all observations selected for this study both in surface waters and the water-column to be representative of the water masses in this POOZ/HNLC region upstream of the Kerguelen Plateau.

Data for 1985-2011 were extracted from the GLODAP data-product, version V2.2021 (Lauvset et al., 2021 a, b; Table S1a). Observations collected during OISO cruises in 2012-2021 (Lo Monaco, 2020; Lo Monaco et al., 2021) will be included in GLODAP-V3. For the surface water properties, all available underway $fCO_2$ data were selected (Figure 1). This includes one cruise in November 1962 (Keeling and Waterman, 1968) and 41 cruises in 1991-2021 (Table S1b). All surface temperature, salinity and $fCO_2$ data were extracted from the SOCAT data-product version v2022 (Surface Ocean $CO_2$ Atlas, Bakker et al., 2016, 2022).




**2.2 Methods**



The methods for surface underway $fCO_2$ and biogeochemical properties (Oxygen, $A_T$-$C_T$, nutrients) in
the water-column for the INDIGO-1, KERFIX and OISO cruises were described in previous studies (e.g.
Poisson et al., 1993; Louanchi et al., 2001; Metzl et al., 2006; Metzl, 2009; Mahieu et al., 2020; Leseurre et al.,
2022). Here we briefly recall the methods for underway $fCO_2$ and water-column observations.

**2.2.1 Surface $fCO_2$ data**

For $fCO_2$ measurements in 1991-2021, sea-surface water was continuously equilibrated with a "thin
film" type equilibrator thermostated with surface seawater (Poisson et al., 1993). The $xCO_2$ in the dried gas was
measured with a non-dispersive infrared analyser (NDIR, Siemens Ultramat 5F or 6F). Standard gases for
calibration (around 270, 350 and 490 ppm) were measured every 6 hours. To correct $xCO_2$ dry measurements to
$fCO_2$ *in situ* data, we used polynomials from Weiss and Price (1980) for vapour pressure and from Copin-
Montégut (1988, 1989) for temperature. Note that when incorporated in the SOCAT data-base, the original $fCO_2$
data are recomputed (Pfeil et al., 2013) using temperature correction from Takahashi et al. (1993). Given the
small difference between equilibrium temperature and sea surface temperature (+0.56 ± 0.30 °C on average for
the cruises in 1998-2021), the $fCO_2$ data from SOCAT used in this analysis (Bakker et al., 2022) are almost
identical (within 1 µatm) to the original $fCO_2$ values from our cruises (www.ncei.noaa.gov/access/ocean-carbon-
data-system/oceans/VOS_Program/OISO.html).

**2.2.2 Water column data**

In 1990-1995, water samples were collected during the KERFIX program on the ship *La Curieuse* at
standard depths using 8 L Niskin bottles mounted on a stainless steel cable and equipped with reversing SIS
pressure and temperature probes. Methods and accuracy for the geochemical measurements used in this analysis
($A_T$, $C_T$, oxygen, nutrients) are detailed by Jeandel et al. (1998) and by Louanchi et al. (2001). From 1998
onwards, the station was occupied within the framework of the OISO long-term monitoring program onboard the
*R.V. Marion-Dufresne*. We used Conductivity-Temperature-Depth (CTD) sensors mounted on a 24 bottles
rosette equipped with 12 L Niskin bottles. Temperature and salinity measurements have an accuracy of 0.002 °C
and 0.005 respectively (Mahieu et al., 2020). Samples for $A_T$ and $C_T$ were filled in 500 mL glass bottles and
poisoned with 100 µL of saturated mercuric chloride solution to halt biological activity. Discrete $C_T$ and $A_T$
samples were analyzed onboard by potentiometric titration derived from the method developed by Edmond
(1970) using a closed cell. Based on replicate samples from the surface or depth, the repeatability for $A_T$ and $C_T$
varies from 1 to 3.5 $\mu mol.kg^{-1}$ depending on the cruise. The accuracy of ±3 $\mu mol.kg^{-1}$ was ensured by daily
analyses of Certified Reference Materials (CRMs) provided by Andrew Dickson's laboratory (Scripps Institute
of Oceanography).
Dissolved oxygen ($O_2$) concentration was determined by a sensor fixed on the rosette and values were
adjusted based on discrete measurements (Winkler method, Carpenter, 1965) using a potentiometric titration
system. Accuracy for $O_2$ is ±2 $\mu mol.kg^{-1}$ (Mahieu et al., 2020). Although long-term deoxygenation in the



Southern ocean has been suggested (Ito et al., 2017; Schmidtko et al., 2017; Oschlies et al., 2018), no significant
trend in $O_2$ was identified over 1985-2021 at this station around 50°S in both surface or in subsurface waters
(e.g. in the layer of the temperature minimum representing winter water for $C_{ant}$ calculations as described later).
However, in the station data a small $O_2$ decrease was detected around 800m in the $O_2$ minimum layer over 36
years (-0.22 ± 0.07 µmol.kg$^{-1}$.yr$^{-1}$). As this has no impact on the interpretation for pH and Ω trends for this
analysis, the observed change of $O_2$ at depth will be not discussed further. Here the $O_2$ data are mainly used for
the calculation of anthropogenic $CO_2$ concentrations and the observed $O_2$ change at depth is too small to have an
impact on temporal variations of $C_{ant}$ concentrations.
Nitrate ($NO_3$) and silicate (DSi) were analyzed on board or at LOCEAN/Paris by colorimetry following
the methods described by Tréguer and Le Corre (1975) for 1998-2008 or from Coverly et al. (2009) for 2009-
2021. The uncertainty of $NO_3$ and DSi measurements is ±0.1 µmol.kg$^{-1}$. Based on replicate measurements for
deep samples, we estimate an error of about 0.3 % for both nutrients. Phosphate ($PO_4$) samples were analyzed in
samples from a few cruises following the method of Murphy and Riley (1962) revised by Strickland and Parsons
(1972) with uncertainty of ± 0.02 µmol.kg$^{-1}$. When nutrient data are not available for a cruise, we used
climatological values based on seasonal nutrients cycles inferred from data from 1990 to 2021. This has a very
small impact on the carbonate system calculations and the trend analysis as we did not detect any significant
trends in nutrients in surface or at depth since 1985 (not shown) as opposed to what has been observed at higher
latitude of the SO (Iida et al., 2013; Hoppema et al., 2015). However, we will see that the inter-annual variability
of nutrients (especially DSi in the HNLC region) might inform on potential changes in biological processes.
For Chlorophyll-a (Chl-a), samples were taken in the top layers (0-150m). One to two liters of seawater
were filtered onto 0.7 µm glass microfiber filters (GF/F, Whatman) and filters were stored at -80°C onboard.
Back at the LOCEAN/Paris laboratory, samples were extracted in 90% acetone (Strickland and Parsons, 1972)
and fluorescence of Chl-a was measured on a Turner Type 450 fluorometer in 1998-2007 and since 2009 at 670
nm on a Hitaschi F-4500 spectrofluorometer (Neveux and Lantoine, 1993).

**2.2.3 Data quality-control and data consistency**

When exploring the trends of ocean properties based on different cruises more than 35 years apart, it is
important to first verify the consistency of the data and if there is any bias or drift. The INDIGO data from 1985
(i.e. prior to CRM available for $A_T$ and $C_T$) were first controlled prior to their incorporation into the original
GLODAP product (Sabine et al., 1999; Key et al., 2004) and corrections for $A_T$ and $C_T$ were revisited within the
framework of the CARINA project (CARbon IN the Atlantic, Lo Monaco et al., 2010). A secondary quality
control was performed on the data from the OISO cruises collected between 1998 and 2011 within the CARINA
and GLODAP-v2 initiatives (Lo Monaco et al., 2010; Olsen et al., 2016). Significant off-sets were identified for
$A_T$-$C_T$ in samples from the KERFIX cruises (1990-1993) compared to INDIGO and OISO data and it was
proposed to correct the original values by -35 µmol.kg$^{-1}$ for $C_T$ and -49 µmol.kg$^{-1}$ for $A_T$ (Metzl et al., 2006).
These corrections were applied in GLODAP version v2.2019 (Olsen et al., 2019) and resulted in coherent $A_T$ and
$C_T$ concentrations for KERFIX in the deep layers compared to other cruises (Supp Mat., Table S2, Figure S1).
The same data quality control protocol as for GLODAP-v2 was applied to data from OISO cruises for the years
2012-2021 (Mahieu et al., 2020). Given the accuracy of the data no systematic bias (excepted in 2014) was
found for the properties measured in 2012-2021. The time-series of $A_T$ and $C_T$ at depth below 1450 m for all





cruises in 1985-2021 show some variability but no trend over 36 years as expected in the bottom waters in this region (Supp. Mat, Figure S1). However, we identified a small bias for $C_T$ in 2014 (cruise OISO-23) where $C_T$ concentrations in the deep water appeared slightly lower (2228-2234 µmol.kg$^{-1}$ in 2014 compared to the mean value of 2240.7 (± 3.7) µmol.kg$^{-1}$, Table S2, Figure S1). When compared to $fCO_2$ in surface waters, we also suspect the $C_T$ data in the mixed-layer in 2014 to be too low by about 10 µmol.kg$^{-1}$ (Figures S2, S3). Therefore we applied a WOCE/GLODAP flag 3 for $C_T$ data of this cruise and will not use the station data in 2014 for the $C_{ant}$ calculations and the trend analysis described in this study.

**2.2.4 CMEMS-LSCE-FFNN model**

As most of the cruises took place during austral summer and data are not available each year, we completed the observations with the results from an ensemble of feed-forward neural network model (CMEMS-LSCE-FFNN or FFNN for simplicity here, Chau et al., 2022). The FFNN model allows mapping at global scale monthly surface $fCO_2$ given SOCAT gridded datasets and ancillary variables. The reconstructed $fCO_2$ is then used to derive monthly surface pH fields as well as air-sea $CO_2$ fluxes. This data product enables to investigate the trends for different seasons and to derive estimates of annual air-sea $CO_2$ fluxes to interpret the change in $CO_2$ uptake, if any. For a full description of the model, access to the data and a statistical evaluation of $fCO_2$ reconstructions please refer to Chau et al. (2022). Within this study, we compared the FFNN $fCO_2$ with observations from 35 cruises for the years between 1991 and 2020 (Table S3, Figure S2a). Excepted for a few periods (January 1993 and January 2002), model-data differences are generally within ± 10 µatm with a mean difference of 2.1 (± 7) µatm for the 35 co-located periods. Note that, as opposed to sea surface $fCO_2$, no temporal trend was identified for the differences between the observed and reconstructed $fCO_2$ (Figure S2b), i.e. the trends of sea surface $fCO_2$ derived from the observations and from the FFNN model should be the same. Aside from the $fCO_2$ reconstructions, surface ocean alkalinity ($A_T$) fields are also provided by using the multivariate linear regression model LIAR (Carter et al., 2016; 2018) based on sea surface temperature, salinity, and nutrient concentration.

**2.2.5 Calculations of carbonate properties**

Based on the data available for each cruise ($fCO_2$, or $A_T$ and $C_T$) or from the FFNN model ($fCO_2$ and $A_T$), other carbonate system properties (pH, [H$^+$], [CO$_3^{2-}$] and Ω) were calculated using the CO2sys program (version CO2sys_v2.5, Orr et al., 2018) developed by Lewis and Wallace (1998) and adapted by Pierrot et al. (2006) with K1 and K2 dissociation constants from Lueker et al. (2000) as recommended (Dickson et al., 2007; Orr et al., 2015; Wanninkhof et al., 2015). The total boron concentration was calculated according to Uppström (1974) and KSO$_4$ from Dickson (1990). To calculate the properties with the underway surface $fCO_2$ dataset, we used the $A_T$/S relationship based on $A_T$-$C_T$ data from OISO cruises in 1998-2019 in the South Indian sector as described by Leseurre et al. (2022):

$A_T$ =64.341 x S +106.764 (rmse = 7.485 µmol.kg$^{-1}$, n = 4775) (Eq. 1)



The use of other $A_T/S$ relationships (e.g. Millero et al., 1998; Jabaud-Jan et al., 2004; Lee et al., 2006;
Carter et al., 2018) would change slightly the $A_T$ concentrations but neither the $A_T$ trend nor the interpretation of
the $C_T$, pH or $\Omega$ trends. However, as salinity is an important predictor in the calculation of $A_T$, $C_T$ or pH from
$fCO_2$ data, we have assessed the original underway salinity data and found biases for few cruises in 1992, 1993
and 1995 (Table S1b). For these cruises or when salinity was not measured we used the salinity from the World
Ocean Atlas, WOA (Antonov et al., 2006) in the SOCAT data-sets (Pfeil et al., 2013, identified "WOA" in Table
S1b). Monthly $fCO_2$ and $A_T$ data extracted from the CMEMS-LSCE-FFNN datasets at the station location
(50.5°S-68.5°E) over 1985-2020 were used to calculate the carbonate properties in the same way as from
observations.

**2.2.6 Comparisons of different datasets and the FFNN model**

To validate the properties calculated using the $fCO_2$ data for 1991-2021 or from the FFNN model over
1985-2020 we compared the calculated values ($A_T$, $C_T$, pH, [$H^+$], [$CO_3^{2-}$], $\Omega$) with those from $A_T$-$C_T$ data
measured in the mixed-layer at stations occupied in 1985 and in 1993-2021. For this comparison, we averaged
the continuous underway $fCO_2$ data selected in a box around the station location (50°S-51.5°S/67.5-69°E, yellow
box in Figure 1). Results of the comparisons between various datasets are detailed in the Supplementary Material
(Tables S3 and S4). During the period 1993-2021, there are 22 stations with co-located $fCO_2$ data for different
seasons (but mainly in summer). Since we found a close agreement between measured $fCO_2$ and the FFNN
model (Table S3, Figure S2), mismatches in all calculated carbonate system properties between the underway
$fCO_2$ dataset and the FFNN model are small, falling within the range of the errors associated with the
calculations (Orr et al., 2018). For example, for 35 co-located periods, the mean differences in calculated $C_T$ of
1.5 (± 5) µmol.kg$^{-1}$ or pH of -0.002 (± 0.008) are in the range of the theoretical error of about 5 µmol.kg$^{-1}$ and
0.007 respectively when taking into account measurements errors on salinity, temperature, nutrients, $fCO_2$ and
$A_T$ (Orr et al., 2018). On the other hand, compared to the station data in the mixed-layer (Table S4), bias for
calculated $A_T$ using Equation 1 is slightly higher by about 5 µmol.kg$^{-1}$. This explains the relatively high
differences for $C_T$ (mean difference around 8 µmol.kg$^{-1}$) and for pH (mean difference around 0.008) calculated
with $fCO_2$ and the $A_T/S$ relationship. The differences of calculated values with observations in 1991-2021 are, on
average, in the range of uncertainties of the carbonate system calculations using $A_T$-$C_T$ pairs (error for $fCO_2$
around 13 µatm and for pH around 0.0144). Importantly, there is no temporal trend for the differences between
calculated and observed properties (Figure S3b). We are thus confident using the selected $fCO_2$ data for the trend
analysis presented in this study. The independent comparison with $A_T$-$C_T$ data at stations also indicates that the
FFNN model results for $A_T$, $C_T$, are close to the observations (Table S4, Table S5, Figure S4) as well as for
calculated pH, [$H^+$], [$CO_3^{2-}$], $\Omega_{Ca}$ and $\Omega_{Ar}$. This somehow validates the use of the FFNN data for the trend
analysis over the period 1985-2020 and for different seasons, although the FFNN model was not constrained by
in-situ $fCO_2$ before 1991 or Chl-a satellite data before 1998. Interestingly, in 1985 the atmospheric $fCO_2$ was
around 335-339 µatm (Dlugokencky and Tans, 2022) and the oceanic $fCO_2$ from the FFNN model was higher
than in the atmosphere from March to October (Figure S4) resulting in an annual $CO_2$ source of +0.8 mol.m$^{-2}$.yr$^{-1}$
in 1985.



## 3 Results and discussion

### 3.1 Variability and trend of sea surface fCO$_2$ and air-sea CO$_2$ fluxes: 1985-2021

The fCO$_2$ observations around 50°S-68°E and their mean values for each cruise are shown in Figure 2a. The fCO$_2$ data in 1991-2021 were available for different seasons but the sampling locations were mainly reoccupied in austral summer (January-February). During austral summer, the ocean fCO$_2$ was generally lower than in the atmosphere (i.e. the ocean was a CO$_2$ sink) whereas in March to October it was near equilibrium. The same distribution is obtained from the FFNN model for 1991-2020 (Figure 2a). The model also indicates that in 1985-1998 the fCO$_2$ during austral winter (May-September) was always higher than the atmospheric fCO$_2$ leading to a CO$_2$ source during this period (Figure 3). The model estimates a decrease of the annual CO$_2$ source in 1985-2001 followed by an increase of the source in 2001-2010 and an increase of the sink in 2010-2020.

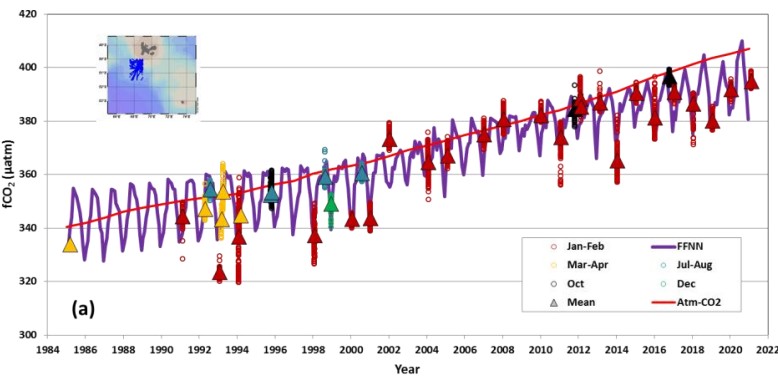

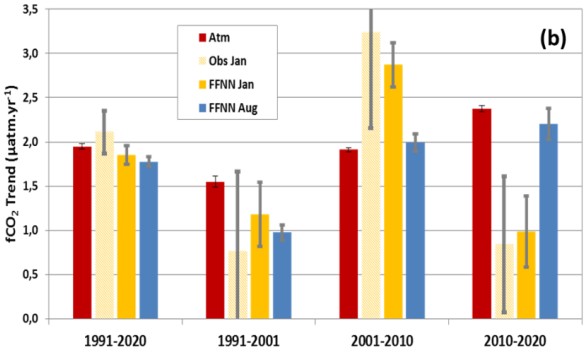

Figure 2: (a): Time-series of sea surface fCO$_2$ observations (µatm) South-West of Kerguelen Island in 1985-2021 (insert map shows the location of observations selected around station at 50°40'S-68°25'E). The color dots correspond to 5 seasons (January-February, March-April, July-August, October and December) and triangles the average for each period. The monthly sea surface fCO$_2$ from the FFNN model is presented for the period 1985-2020 (purple line) and the atmospheric fCO$_2$ represented by red line. In March 1985 there were no underway fCO$_2$ observations and the triangle corresponds to fCO$_2$ calculated with A$_T$-C$_T$ data in the mixed-layer. (b): Trends of atmospheric and oceanic fCO$_2$ (µatm.yr$^{-1}$) for different season and periods based on observations (January) and the FFNN model (January or August).



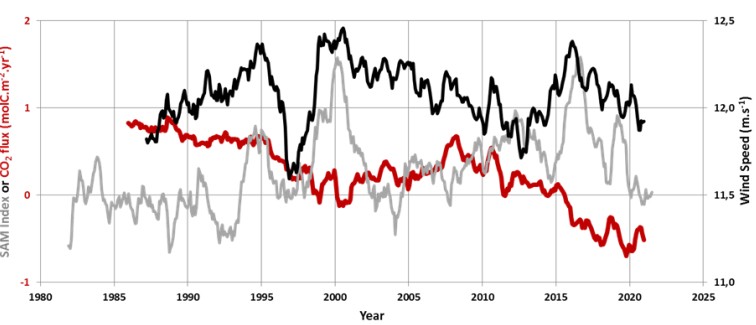

Figure 3: Time series of the SAM index (in grey) in the Southern Ocean, wind-speed (in black, m.s$^{-1}$) and air-sea $CO_2$ flux (molC.m$^{-2}$.yr$^{-1}$) from the FFNN model (in red) at location 50.5°S-68.5°E. Positive (negative) flux represents $CO_2$ source (sink). Wind-speed and SAM are presented for respectively 12-months and 24-months running mean based on monthly values. Note the positive SAM in 1998-2003 and 2010-2020. SAM data from Marshall (2003), http://www.nerc-bas.ac.uk/icd/gjma/sam.html, last access 14/8/2021. Wind speed data from ERA5 (Hersbach et al., 2020).

For the last cruise in February 2021, the average $fCO_2$ was 394.9 (± 1.5) µatm (Figure 2a), about 10 µatm lower than in the atmosphere (a small $CO_2$ sink). This is +50.5 µatm higher than $fCO_2$ observed during the first cruise in February 1991 ($fCO_2$ = 344.4 ± 5.2 µatm). During the same period, the atmospheric $CO_2$ increased from 354 ppm in 1991 to 411 ppm in 2021 in this region (recorded at Crozet Island, Dlugokencky and Tans, 2022). This first comparison of two cruises 30 years apart indicates that the ocean $fCO_2$ increased at a rate (+1.7 µatm.yr$^{-1}$) close to that of the atmosphere (+1.9 µatm.yr$^{-1}$). During the same period, we observed some variations in $A_T$ (average $A_T$ = 2276.5 ± 4.5 µmol.kg$^{-1}$) and a clear increase in $C_T$ (Figure 4a and S5).

The $C_T$ concentration in the mixed-layer in summer 2021 was 2134.0 (± 1.8) µmol.kg$^{-1}$, much higher than in summer 1993 ($C_T$ = 2115.8 ± 2.6 µmol.kg$^{-1}$). The difference over 28 years of +22.1 µmol.kg$^{-1}$ corresponds to an annual $C_T$ increase of +0.8 µmol.kg$^{-1}$.yr$^{-1}$. At constant temperature and $A_T$, this would translate in an increase of oceanic $fCO_2$ of +1.9 µatm.yr$^{-1}$, i.e. equal to the atmospheric rate. The same comparison for October shows that $fCO_2$ in 2016 was +43.8 µatm higher compared to 1995 (Figure 2a), i.e. a rate of +2.1 µatm.yr$^{-1}$. The $C_T$ concentrations in October 2016 were also much higher than in 1993 (Figure 4a and S5). Over 23 years the observed $C_T$ increase in October (+22.6 µmol.kg$^{-1}$) corresponds to a rate of +0.98 µmol.kg$^{-1}$.yr$^{-1}$ that is faster than the rate of +0.8 µmol.kg$^{-1}$.yr$^{-1}$ derived from summer data in 2021 and 1993. At constant $A_T$ this would translate in an increase of oceanic $fCO_2$ of +2.5 µatm.yr$^{-1}$ in October, higher than the trend of +2.1 µatm.yr$^{-1}$ computed from $fCO_2$ data. Part of the difference may be explained by $A_T$ that was slightly higher (+6 µmol.kg$^{-1}$) in October 2016 compared to 1993 (Figure S5).

Given the temporal variability of observed $C_T$ in summer and the evolution of the annual air-sea $CO_2$ flux (Figure 3), decadal $fCO_2$ and pH trends as well as associated drivers need to be analyzed for different seasons and periods. This approach allows exploring links with the variability of primary production and/or the Southern Annual Mode (SAM). Shifts from a positive to a negative SAM index (Figure 3) will strengthen upwelling and impact ocean properties throughout the water column including $C_T$, nutrients, primary production or pH (e.g. Lovenduski and Gruber, 2005; Lenton et al., 2009; Hauck et al., 2013; Hoppema et al., 2015; Pardo et al., 2017).




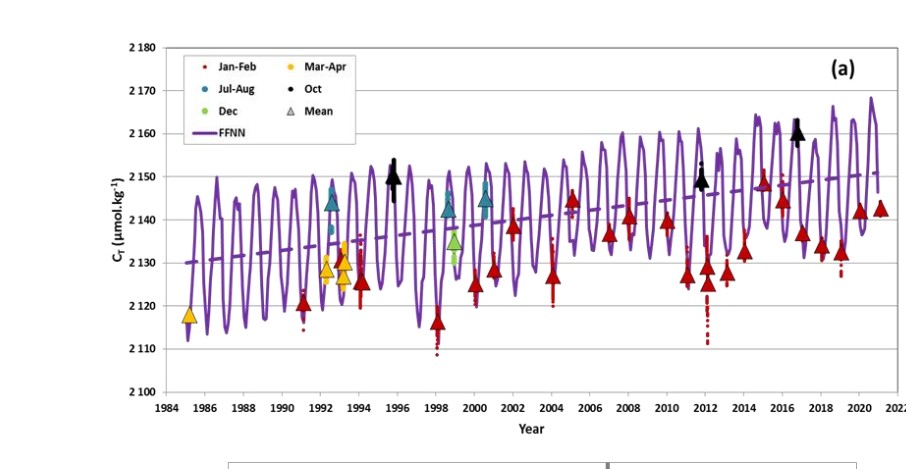

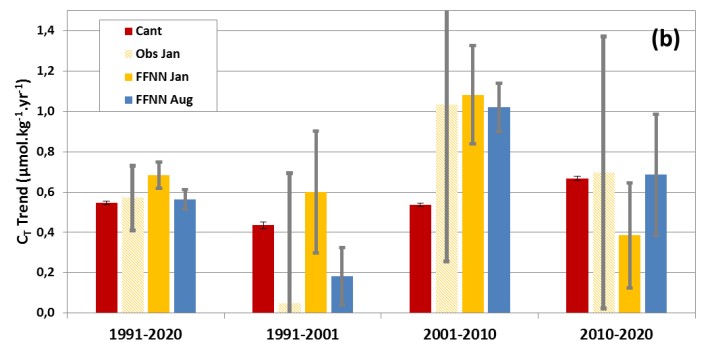

Figure 4: (a): Time-series of surface $C_T$ (µmol.kg$^{-1}$) around station at 50°40'S-68°25'E calculated from fCO$_2$ data (Figure 2) using the $A_T$/S relation (see text). The color dots correspond to 5 seasons (January-February, March-April, July-August, October and December) and triangles the average for each cruise. The monthly sea surface $C_T$ from the FFNN model is presented for the period 1985-2020 (purple line). The annual $C_T$ trend of +0.58 ±0.05 µmol.kg$^{-1}$.yr$^{-1}$ (dashed line) is derived from the FFNN monthly data. In March 1985 the triangle corresponds to the observed $C_T$ in the mixed-layer. (b): Trends of sea surface $C_T$ (µmol.kg$^{-1}$.yr$^{-1}$) for different season and periods based on observations (for January) and the FFNN model (for January or August). The trend for $C_{ant}$ (µmol.kg$^{-1}$.yr$^{-1}$) is also shown (red bars) based on estimates in the winter water.

Summer data are characterized by a strong inter-annual variability between 1991-2021 (Figures 2a and 4a) with the ocean being a CO$_2$ source in January 2002, but a strong sink in January 1993, 1998, 2014, 2016 and 2019. In January 1998, when the surface ocean experienced a warm anomaly (Jabaud-Jan et al., 2004), the low fCO$_2$ of 337 µatm and the low $C_T$ of 2110 µmol.kg$^{-1}$ (Figure 4a and S5) co-occurred with intense primary production, probably supported by diatoms as suggested by very low DSi concentrations (< 2 µmol.kg$^{-1}$ down to 100m, Figure S6). In January 2014 and 2016, mixed-layer DSi concentrations were also remarkably small (< 5 µmol.kg$^{-1}$ down to 75m, Figure S6). In 2014 low DSi coincided with Chl-a levels that started to increase in mid-November 2013 and stayed at high level until February 2014 (Surface Chl-a > 0.3 mg.m$^{-3}$, Figures 5 and S7). The intense primary production contributed to the low fCO$_2$ of 365 µatm reached by mid-January 2014, a value as low as in 2004 (Figure 2a). To the contrary, in 2002 relatively low Chl-a (mean Chl-a < 0.2 mg.m$^{-3}$, Figure 5) was associated with higher levels of fCO$_2$ (373 µatm), $C_T$ (2128 µmol.kg$^{-1}$, Figure 4a, Figure S5a) and DSi (Figure S6). This was also associated with higher salinity indicative of entrainment that might be related to storm





events that would occurred few days before the measurements leading to brief positive $fCO_2$ anomaly as recently
observed from Glider data in the subpolar South Atlantic (Nicholson et al., 2022). As opposed to the other
periods the ocean was a source of $CO_2$ in summer 2002 (this particular year was not well reconstructed by the
FFNN model, Figure 2a and Figure S2b). The important inter-annual variability observed in summer indicates
that in this region historically referred to as HNLC (Minas and Minas, 1992), primary production could
significantly impact $fCO_2$ level in summer (Jabaud-Jan et al., 2004; Pasquer et al., 2015; Gregor et al., 2018), a
result that needs to be taken into account when evaluating drivers of inter-annual variability (Rustogi et al.,
2023) and the decadal trends of $fCO_2$ or pH.
The Chl-a time-series derived from MODIS suggests higher concentrations in recent years with Chl-a
peaks identified in 2014, 2016, 2018, 2019 and 2021 (Figure 5 and S7) when the oceanic $fCO_2$ in summer was
well below the atmospheric level (Figure 2a).

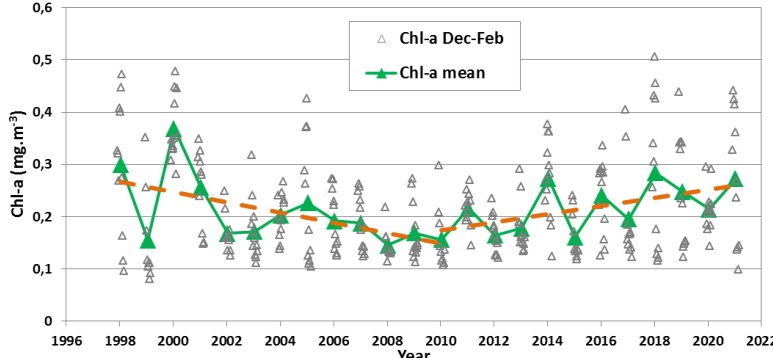

Figure 5: Time-series (1998-2021) of sea surface Chl-a (mg.m$^{-3}$) in summer (December-February) from weekly satellite data
(SeaWIFS and MODIS, triangles) and associated mean (green triangles). The trends in 1998-2010 and 2010-2021 of
respectively -0.0099 (± 0.0041) and +0.0078 (± 0.0032) mg.m$^{-3}$.yr$^{-1}$ (dashed orange) indicate a decrease or increase of the
primary production that drives part of the $fCO_2$ and $C_T$ stability observed in the recent period (Figure 2, Figure 4). The full
Chl-a record is shown in Supp. Mat. Figure S7.
The primary production lowers $C_T$ concentrations and $fCO_2$, i.e. opposite to the $C_T$ increase from
anthropogenic $CO_2$ uptake. These counteracting processes might explain the relatively stable $fCO_2$ previously
observed in the Indian POOZ in summer 2007-2019 with an annual $fCO_2$ rate of increase of only +0.3 (± 0.2)
µatm.yr$^{-1}$ (Leseurre et al., 2022). This low rate is confirmed here with the new data obtained in 2020-2021
(Figure 2b and Figure S8). For the period 2010-2021, the oceanic $fCO_2$ trend in summer derived from
observations and the FFNN model is lower than +1 µatm.yr$^{-1}$ (Table 1), i.e. much lower than the atmospheric
$fCO_2$ rate of +2.4 µatm.yr$^{-1}$ and the oceanic $fCO_2$ trend of +2.21 (± 0.17) µatm.yr$^{-1}$ in winter (Figure 2b). This
rate is also lower compared to the change observed in October (+2.9 µatm.yr$^{-1}$) albeit being only based on 2
cruises in 2011 and 2016 (Figure 2a). As the low $fCO_2$ trend in recent years is detected for summer only this is
likely linked to an increase in primary production, as suggested by Chl-a records (Figure 5). In 1998-2010 the
summer Chl-a concentrations decreased at a rate of -0.099 (± 0.041) mg.m$^{-3}$.decade$^{-1}$ whereas in 2010-2021 Chl-
a increased by +0.078 (± 0.032) mg.m$^{-3}$.decade$^{-1}$ (Figure 5). These trends are coherent with previous studies, e.g.
the reduced net primary productivity reported in the Indian Antarctic zone in 1997-2007 (e.g. Arrigo et al., 2008;
Takao et al., 2012) and the shift of the Chl-a trend at large scale in the HNLC region of the Southern Ocean in
2010 (Basterretxea et al., 2023). As a consequence, after 2010 the difference between oceanic and atmospheric





$fCO_2$, $\Delta fCO_2$ (where $\Delta fCO_2 = fCO_2^{oce}-fCO_2^{atm}$) decreased in summer (-1.4 µatm.yr$^{-1}$) and the annual $CO_2$ flux
progressively varied from a source of +0.45 molC.m$^{-2}$.yr$^{-1}$ in 2010 to a sink of -0.63 molC.m$^{-2}$.yr$^{-1}$ in 2020
(Figure 3). In addition, because the wind speed was stable during this period (12.0 ± 0.9 m.s$^{-1}$ on average in
2010-2020, Figure 3), the variation of the air-sea $CO_2$ flux was mainly controlled by $\Delta fCO_2$ (e.g. Gu et al., 2023)
and the decadal variation of primary production imprinted a significant change on the $fCO_2$ trend and air-sea
$CO_2$ flux in this HNLC region. In the region investigated here, increasing Chl-a levels co-occurred with shifts of
the SAM index to a positive state (Figure 3), a link previously suggested south of the Polar Front in the SO but
for a short period 1997-2004 (Lovenduski and Gruber, 2005).
Another process to take into account for interpreting $fCO_2$ trends is the change in temperature in surface
waters. Previous analysis suggested a progressive warming in the region investigated here (Auger et al., 2021 for
summer 1993-2017). For 1998-2019 Leseurre et al. (2022) estimated a warming of Indian POOZ surface waters
of +0.03 (± 0.02) °C.yr$^{-1}$. Extending the time-series for the period 1991-2021 (Figure S9a) we note that the
surface temperature presents sub-decadal variability and that the ocean cooled after 2018 with a trend of -0.474
(± 0.164) °C.yr$^{-1}$ in 2018-2021 based on the monthly sea surface temperature (SST, Figure S9b). The trend
derived from our in-situ observations in summer 2018-2021 was -0.253 (± 0.092) °C.yr$^{-1}$.
In 2019, the lower temperature and relatively high Chl-a lead to low $fCO_2$ (380 µatm, Figure 2a) and
low $C_T$ (2128 µmol.kg$^{-1}$) compared to 2018 ($fCO_2$ = 386 µatm; $C_T$ = 2137 µmol.kg$^{-1}$, Figure 4a). The decrease in
observed $fCO_2$ from summer 2018 to 2019, also reconstructed by the FFNN model (Figure 2a), is contrary to the
expected $fCO_2$ and $C_T$ increase due to anthropogenic uptake. In 2020, although the temperature was also lower,
the oceanic $fCO_2$ was higher (392 µatm) probably due to lower primary production as suggested by higher DSi
(Figure S6), as well as from $C_T$ (2135 µmol.kg$^{-1}$, Figure 4a) and Chl-a records (Figure 5). In January 2021 the
temperature was close to that in January 2020, and both $fCO_2$ and $C_T$ were slightly higher (395 µatm, 2139
µmol.kg$^{-1}$). $A_T$ concentrations were stable between 2018 and 2021 (2278.9 ± 1.8 µmol.kg$^{-1}$, Figure S5) indicating
no effect of $A_T$ on the observed $fCO_2$ change in this region as opposed to the areas north of the Polar Front in the
Indian Ocean where $A_T$ variations are often linked to coccolithophores blooms (Balch et al., 2016; Smith et al.,

2017).

The inter-annual variability observed in 1991-2021 highlights the competitive processes that drive $C_T$,
$fCO_2$ or pH temporal variations. In summer 2018-2019, cooling and increased primary production both lead to
low $fCO_2$ counteracting the effect of anthropogenic $CO_2$ uptake. Given the changes of Chl-a, SST and air-sea
$CO_2$ flux, trends will be evaluated for three periods, 1991-2001, 2001-2010 and 2010-2020. In order to separate
natural and anthropogenic contributions, the anthropogenic $CO_2$ signal is estimated in the following section.

**3.2 Anthropogenic $CO_2$**

**3.2.1 Anthropogenic $CO_2$ in the water column**

To calculate anthropogenic $CO_2$ concentrations ($C_{ant}$), we used the TrOCA method developed by
Touratier et al. (2007) and previously applied in the southern Indian Ocean (Mahieu et al., 2020; Leseurre et al.,
2022). Such an indirect method is not suitable for evaluating $C_{ant}$ concentrations in surface waters due to
biological activity and gas exchange and we restrict the $C_{ant}$ calculations below the productive layer around





150m. In the region south of the Polar Front, a well-defined subsurface temperature minimum is observed each
year characterizing the winter water (WW) (Figure 6a).

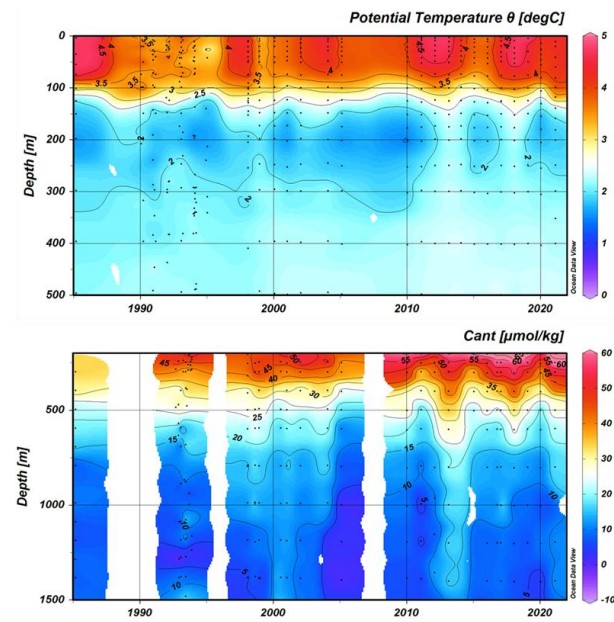

Figure 6: Hovmoller section (Depth-Time) of potential temperature (°C) and anthropogenic $CO_2$ ($C_{ant}$, µmol.kg$^{-1}$) in 1985-2021 at station OISO-KERFIX (50°40'S-68°25'E). The section for temperature is presented in the layer 0-500m and for summer to highlight the temperature minimum around 200m (winter water, WW). The section for $C_{ant}$ is limited below 200m. Section produced with ODV (Schlitzer, 2018).

The $C_T$ and $C_{ant}$ concentrations increased over time in the water column, a signal that is most
pronounced in the top layers (0-400m, Figure 6b). In the deep layer, the presence of the Indo-Pacific Deep Water
(IPDW) around 600-800m is identified by a maximum of $C_T$ ($C_T$ > 2250 µmol.kg$^{-1}$) and a minimum of $O_2$ ($O_2$
close to or < 180 µmol.kg$^{-1}$, Figure S10) (Talley, 2013; Chen et al., 2022). In the IPDW layer restricted to the
neutral density (ND) range 27.75-27.85 kg.m$^{-3}$ there is no significant change in $C_T$ over time (Figure S10). In
that layer the $C_{ant}$ concentrations in 1985 (17.3 µmol.kg$^{-1}$) were almost identical to those evaluated in 2021 (21.2
µmol.kg$^{-1}$), considering the uncertainty in the $C_{ant}$ calculations (± 6.5 µmol.kg$^{-1}$, Touratier et al., 2007). As
discussed above (section 2.2.3) the $C_T$ and $A_T$ concentrations in the bottom layer (>1450m) were stable in 1985-
2021 (Table S2, Figure S1). Below 800m, the $C_{ant}$ concentrations were small but not null (Figure 6b). The
average $C_{ant}$ concentration below 800m for all years and seasons is 7.97 (± 5.31) µmol.kg$^{-1}$ (n=123) with a very
small change detected over time ($C_{ant}$ = 7.73 ± 1.27 µmol.kg$^{-1}$ in 1985 and $C_{ant}$= 10.45 ± 0.62 µmol.kg$^{-1}$ in 2021).

**3.2.2 Anthropogenic $CO_2$ trend in the subsurface**

To separate the natural and anthropogenic signals in surface waters for the driver analysis we assume
that $C_{ant}$ in the WW is representative of $C_{ant}$ in the mixed-layer (ML). This is confirmed with few stations
occupied during winter showing that $C_{ant}$ concentrations in the WW in summer are almost equal to $C_{ant}$ in the



ML during the preceding winter (Figure S11). The variation of $C_{ant}$ in the WW for 1985-2021 is presented in
Figure 7a for all seasons. In 1985 the $C_{ant}$ concentration in the WW was 47.1 µmol.kg$^{-1}$ and $C_{ant}$ reached a
maximum of 71.7 µmol.kg$^{-1}$ in 2021. The data selected at 200m present some inter-annual variability like the
relatively low $C_{ant}$ in 1998, 2005 or 2020 probably related to natural variability. In 1998 and in 2020 the $O_2$
concentrations were slightly lower in the WW (< 300 µmol.kg$^{-1}$) explaining the lower $C_{ant}$ concentration (44.8
µmol.kg$^{-1}$ in 1998 and 53.8 µmol.kg$^{-1}$ in 2020). In 2005 anomalies of $C_T$, $O_2$ and temperature concur to explain
the lower $C_{ant}$ (43.9 µmol.kg$^{-1}$).

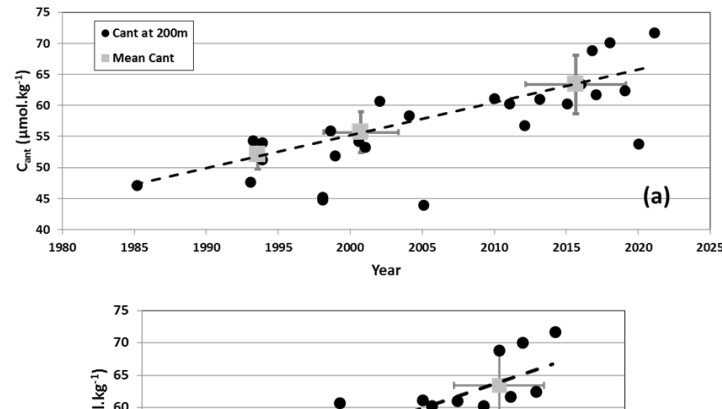

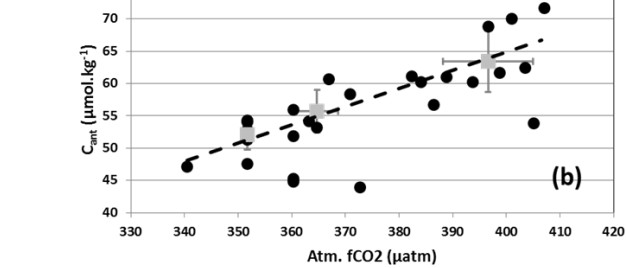

Figure 7: (a): Time-series of anthropogenic $CO_2$ ($C_{ant}$ µmol.kg$^{-1}$) estimated in the winter water layer (WW around 200m, see
figure 6) in 1985-2021 at station OISO-KERFIX (50°40'S-68°25'E). Black dots are the individual data in the WW and the
grey squares the average for the 1990s, 2000s and 2010s (anomalies in 1998, 2005 and 2020 filtered). The $C_{ant}$ trend of +0.53
(± 0.01) µmol.kg$^{-1}$.yr$^{-1}$ is represented (dashed line). (b): same data for $C_{ant}$ versus atmospheric f$CO_2$ (the slope is +0.263 ±
0.042 µmol.kg$^{-1}$.µatm$^{-1}$).
From 1985 to 2021, we estimate a $C_{ant}$ trend in WW of +0.49 (± 0.09) µmol.kg$^{-1}$.yr$^{-1}$. When the $C_{ant}$
anomalies in 1998, 2005 and 2020 are discarded, this $C_{ant}$ trend is +0.53 (± 0.01) µmol.kg$^{-1}$.yr$^{-1}$ (Figure 7a). As
expected, the $C_{ant}$ concentrations in the ocean are positively related to atmospheric $CO_2$ (slope +0.263 ± 0.042
µmol.kg$^{-1}$.µatm$^{-1}$, Figure 7b). Interestingly the slope observed south of the PF in the Indian Ocean is close to that
observed in the Antarctic Intermediate waters (AAIW) in the South Atlantic (+0.23 ± 0.05 µmol.kg$^{-1}$.µatm$^{-1}$,
Fontela et al., 2021). At large scale, Gruber et al. (2019 a, b) evaluated $C_{ant}$ changes between 1994 and 2007 in
the global ocean. In the South Indian sector, they estimated a mean $C_{ant}$ accumulation in the surface of +6.0 (±
1.1) µmol kg$^{-1}$ in the band 50-55°S south of the PF. At our station location (50-52°S/68°E) in the layer 0-250m,
the $C_{ant}$ accumulated from 1994 to 2007 was +5.67 (± 1.47) µmol kg$^{-1}$. In 13 years, this corresponds to a trend of
+0.44 (± 0.11) µmol.kg$^{-1}$.yr$^{-1}$. Gruber et al. (2019 a, b) did not use the data presented here allowing for an
independent comparison to the present study. Estimates of $C_{ant}$ accumulation by Gruber et al. (2019 a, b) are in
agreement with ours for 1991-2008 (+0.46 µmol.kg$^{-1}$.yr$^{-1}$) but lower than reported here between 2008 and 2021
(+0.61 µmol.kg$^{-1}$.yr$^{-1}$).



### 3.2.3 Anthropogenic and surface $C_T$ seasonal trends

The $C_{ant}$ trend in WW over 1985-2021 ($+0.53 \pm 0.01$ µmol.kg$^{-1}$.yr$^{-1}$) is slightly lower than the annual $C_T$ trend in surface derived from the FFNN model for 1985-2020 ($C_T$ trend = $+0.58 \pm 0.05$ µmol.kg$^{-1}$.yr$^{-1}$ Figure 4a) suggesting that anthropogenic $CO_2$ uptake explains 86% of the $C_T$ increase in surface. In 1991-2020 the surface $C_T$ trend appears slightly higher in January ($+0.68 \pm 0.07$ µmol.kg$^{-1}$.yr$^{-1}$) than in August ($+0.56 \pm 0.04$ µmol.kg$^{-1}$.yr$^{-1}$, Figure 4b). This suggests that in addition to the increase of $C_T$ due to anthropogenic $CO_2$ other processes count such as the variability of the biological activity, vertical mixing or upwelling. Indeed, as for $fCO_2$ (Figures 2b), the $C_T$ growth rate also depends on seasons and decades (Figure 4b). In 1991-2001 the $C_T$ trend from the observations ($+0.05 \pm 0.64$ µmol.kg$^{-1}$.yr$^{-1}$) is highly uncertain due to few data and the large variability (Figures 4a, b). The FFNN model showed that the $C_T$ trend in summer was faster than in winter and the winter $C_T$ trend lower than the $C_{ant}$ trend in subsurface (Figure 4b). This is because during that decade, the higher primary production in 1998 created a negative $C_T$ anomaly (Figure 4a) not compensated by the accumulation of $C_{ant}$.

In 2001-2010 the $C_T$ trends were much faster than in 1991-2001 and they were the same for both seasons (around 1 µmol.kg$^{-1}$.yr$^{-1}$, Figure 4b). For this decade the summer $C_T$ trends from the observations and the FFNN model are coherent. They were also twice the $C_{ant}$ rate in WW that could be explained by enhanced upwelling of $C_T$-rich deep waters during this period after the SAM reached a high positive index (Figure 3; Lenton and Matear, 2007; Le Quéré et al., 2007; Hauck et al., 2013). However, in 2001-2010 we did not detect any clear change at depth for ocean properties (except for $C_T$ and $C_{ant}$) that would support this assumption (enhanced upwelling). The rapid $C_T$ (and $fCO_2$) trend for this decade is probably due to processes at the surface rather than changes in the water column (vertical mixing or upwelling). In 2010-2020 $C_T$ trends are lower than in 2001-2010 (Figure 4b). For summer, this is identified from both observations and the FFNN model. In winter the $C_T$ trend is close to $C_{ant}$ indicative of the anthropogenic accumulation. The low $C_T$ trend at the surface in summer, about half the $C_{ant}$ trend for the FFNN model, is likely due to the increase of primary production after 2010 as described above (Figure 5). It appears thus that the impact of biological activity and its variability in summer could counteract that of anthropogenic $CO_2$ and explain the temporal change of the carbonate system in surface.

Given the differences of the $fCO_2$ and $C_T$ trends in summer and winter (Figures 2b and 4b) we explored the temporal variations of the seasonality. For each year we estimated the differences between August and January (Figure 8a). The seasonal amplitude for $C_T$ was on average 26.1 ($\pm$ 3.4) µmol.kg$^{-1}$ and for $fCO_2$ 15.1 ($\pm$ 5.6). Interestingly, the $fCO_2$ seasonal amplitude reached a minimum around 2008-2010 and increased over 2010-2020. This signal appears correlated with surface Chl-a in summer (Figure 8b). The inter-annual variability of the seasonality is clearly identified when comparing $C_T$ with $C_T$ calculated due only to $C_{ant}$ accumulation after 2010 (Figure S12c). This supports the conclusion that in addition to the $C_{ant}$ accumulation, the variations of phytoplanktonic biomass imprinted inter-annual variability on $C_T$ and $fCO_2$ in summer. This holds for the seasonal amplitude as the results for winter follows the $C_{ant}$ trend (Figure 4b, Figure S12a). The same is true for pH for which reduced seasonal amplitude was found when the production was low (not shown). However, over 36 years (1985-2020) we did not identify a long-term trend of the seasonal amplitude for $C_T$ or for $fCO_2$ as suggested by other studies (Landschützer et al., 2018; Rodgers et al., 2023; Shadwick et al 2023). Our results highlight a variability over 5-10 years (Figure 8a) and suggest a potential change in seasonality and annual $CO_2$





sink if primary production changes in the future (e.g. Bopp et al., 2013; Leung et al., 2015; Fu et al., 2016;

Kwiatkowski et al., 2020; Krumhardt et al., 2022; Seifert et al., 2023).

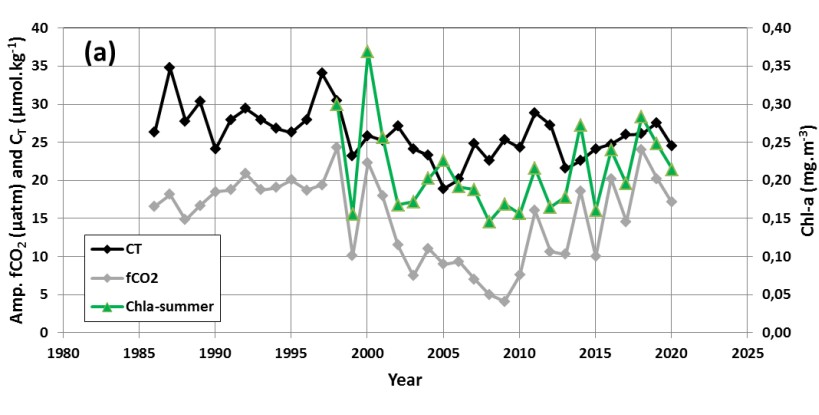

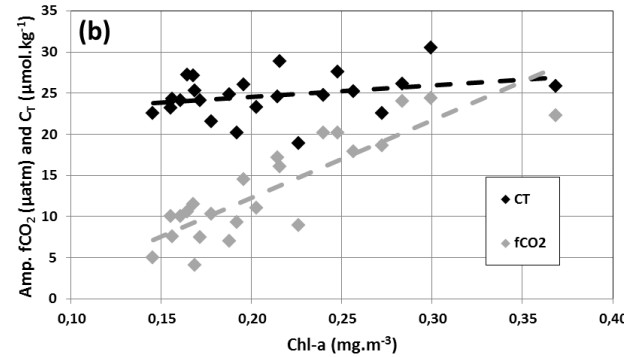

Figure 8: (a): Time-series of the seasonal amplitude (August minus January) for surface $C_T$ (black, µmol.kg$^{-1}$) and $fCO_2$
(grey, µam) from the FFNN model at station OISO-KERFIX (50°40'S-68°25'E). Also shown are the mean surface Chl-a
(green, mg.m$^{-3}$) in summer in 1998-2021. (b): Seasonal amplitude of $fCO_2$ and $C_T$ versus summer Chl-a for 1998-2020. The
dashed lines indicate that the seasonal amplitude (August-January) increases when Chl-a is higher.

**3.3 Anthropogenic $CO_2$ drives acidification in surface and the water column**

**3.3.1 Surface pH trend**

To explore the temporal change of pH in surface water we used the $fCO_2$ observations and the monthly

results from the FFNN model. For both data-sets pH was calculated from $fCO_2$ and $A_T$ reconstructed as
described in section 2.2.5. Figure 9a presents the time-series of pH in the surface (the same time-series for [H$^+$]
concentrations is shown in Figure S13). For the full period, 1985-2020, the annual pH trend derived from the
FFNN model is -0.0165.decade$^{-1}$ (± 0.0004) exactly the same as derived at large scale in the Southern Ocean
(south of 44°S) for the period 1993-2018 (Iida et al., 2021, Table 1) but when restricted to this period, 1993-
2018, the trend from the FFNN model appears slightly faster of -0.0182.decade$^{-1}$ (± 0.0006). This is less than the
pH trend of -0.020 (± 0.002).decade$^{-1}$ derived from pCO$_2$ data in the SO SubPolar Seasonally Stratified biome
around 40-50°S (SO-SPSS) for 1981-2011 (Table 1, Lauvset et al., 2015) and close to the pH trend of -0.0189 (±





0.0010).decade$^{-1}$ based on OceanSODA-ETH reconstructed fields in the SO-SPSS for the period 1982-2021 (Ma
et al., 2023). However, as for fCO$_2$ and C$_T$, different pH trends were estimated in summer and winter as well as
depending on the periods (Figure 9b, Table 1).

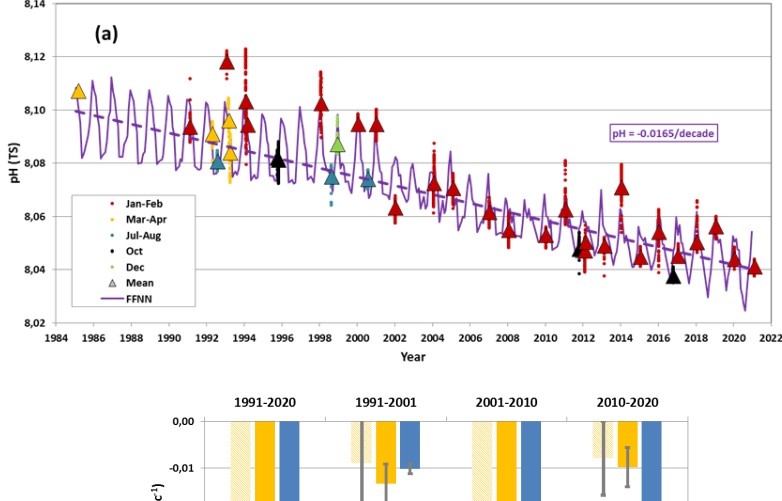

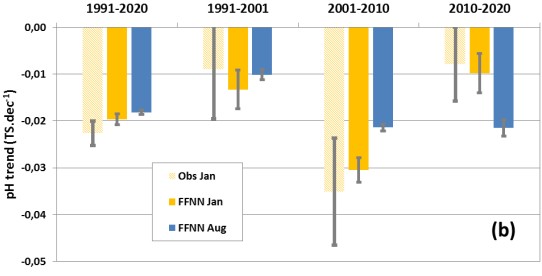

Figure 9: (a): Time-series of surface pH (TS) around station OISO-KERFIX (50°40'S-68°25'E) calculated from fCO$_2$ data
(Figure 2) using the A$_T$/S relation (see text). The color dots correspond to 5 seasons (January-February, March-April, July-
August, October and December) and triangles the average for each cruise. The monthly sea surface pH from the FFNN model
is presented for the period 1985-2020 (purple line). The annual pH trend in 1985-2020 of -0.0165.decade$^{-1}$ (± 0.0004)
(dashed line purple) is derived from the FFNN monthly data (the same figure for [H$^+$] concentrations is presented in Supp.
Mat. Figure S13). (b): Trends of pH (TS.decade$^{-1}$) for different seasons and periods based on observations (January) and the
FFNN model (January or August).

The winter pH decreased was faster in recent years, mirroring the winter fCO$_2$ trend (Figure 2b). On the
opposite, in summer, the pH trend presents a large variability at decadal scale and was lower in 2010-2020. In
summer 2001-2010, the pH trend from the FFNN model was -0.0304.decade$^{-1}$ (± 0.0026) whereas in 2010-2020,
it was -0.0098.decade$^{-1}$ (± 0.0042) (Figure 9b, Table 1). Although the trends based on the observations are less
robust because the cruises were not conducted each year the reduced pH trend in summer after 2010 is confirmed
from in-situ data (-0.0351 ± 0.0114 .decade$^{-1}$ in 2001-2010 against -0.0078 ± 0.0079 .decade$^{-1}$ in 2010-2020,
Figure 9b, Table 1). These results show that the pH trend varied significantly from decade to decade and that the
decrease of pH since 1985 was mainly driven by anthropogenic CO$_2$. This is revealed in the winter water when
comparing pH and pre-industrial pH (Figure 10a). Here, the pre-industrial pH (pH-PI) was calculated after
subtracting C$_{ant}$ values from the observed C$_T$ concentrations for each sample in the WW layer. Interestingly the
pH trend in the WW of -0.0161 (± 0.0033).decade$^{-1}$ (here deduced from the station A$_T$-C$_T$ data in 1985-2021) is
very close to the long-term trend in surface from the FFNN model in 1985-2020 (-0.0165.decade$^{-1}$ ± 0.0004).
This trend is slightly faster than the pH trends of -0.0134 (± 0.001).decade$^{-1}$ recently estimated in subsurface





waters (100-210m) of the Southern Ocean south of the PF and derived for years 1994-2017 from historical data
and BGC-Argo floats (Mazloff et al., 2023). For the same period, 1994-2017, at the OISO-KERFIX station we
estimate a pH trend in the WW of -0.0168 (± 0.0043).decade$^{-1}$ and of -0.0186 (± 0.0006) .decade$^{-1}$ in surface
waters from the FFNN model.

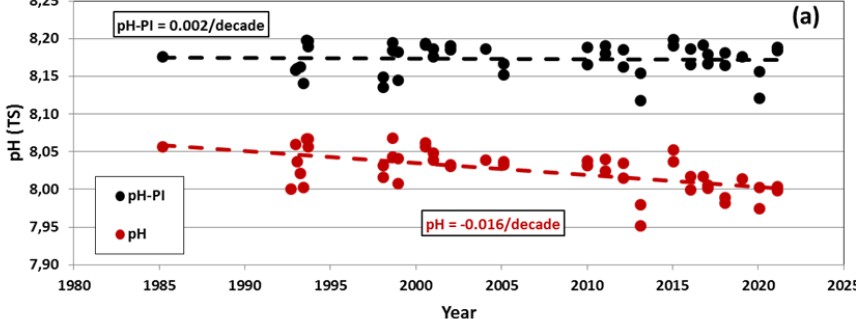

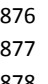

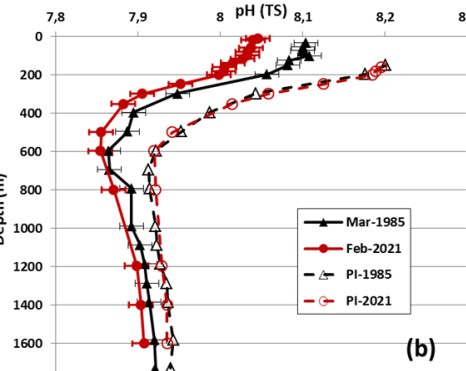

Figure 10: (a): Time-series of pH (red dots) and pre-industrial pH (pH-PI, black dots) estimated in the winter water layer (WW around 200m, see figure 6) in 1985-2021 at station OISO-KERFIX (50°40'S-68°25'E). pH-PI for each sample was calculated after subtracting C$_{ant}$ to C$_T$. The pH trend from the present days is -0.0161 (± 0.0033).decade$^{-1}$ (red dashed line). No trend is observed for pH-PI (black dashed). The mean pH-PI in the WW is 8.173 (± 0.020, n =45). (b): Profiles of pH and pH-PI evaluated from March 1985 (black symbols) and February 2021 data (red symbols). The profiles for pH-PI are shown below 150m only as C$_{ant}$ estimates are not available in surface layer. Note that the pH-PI profiles are the same either using 1985 or 2021 data.

A for other properties (A$_T$, O$_2$, temperature, salinity and nutrients), the pre-industrial pH (pH-PI) does

not change over time in the WW (mean pH-PI = 8.173 ± 0.020, n=45, Figure 10a). The pH-PI in the WW is in
the range of the pre-industrial surface pH value in the Southern Ocean (8.2 for year 1750 and 8.18 for year 1850)
derived from Earth system Models (Jiang et al., 2023, their Table S9). In the WW at our location the modern pH
(1985-2021) was on average -0.147 (± 0.021) lower than pre-industrial pH. In 1985 pH in the WW was -0.119
lower than pH-PI and in 2021 it was -0.184 lower than pH-PI (Figure 10a). The progressive decrease of pH was
clearly linked to C$_{ant}$ concentrations in the WW layer and the pH decrease identified below that layer in the water
column (Figure 10b).



### 3.3.2 Temporal change in the water column

From 1985 to 2021, signals of decreasing pH and increasing $C_T$ in surface waters are propagated in the water column down to about 500m. As mentioned above the data in 1985 (first occupation of the station) reveal significant $C_{ant}$ levels across the water column (Figure 6b). Therefore the pH down to 1400m was already lower in 1985 than at pre-industrial times (Figure 10b). However, the largest $C_{ant}$ increases were found in the top layers and changes in pH from 1985 to 2021 were small below 500m (Figure 10b, Figure S14). While observations for all years fall on a common linear relationship between $C_{ant}$ and $pH_{ant}$ for depths greater than 500 m, the change in pH for a given level of $C_{ant}$ increases with time for layers shallower than 500 m (Figure 11).

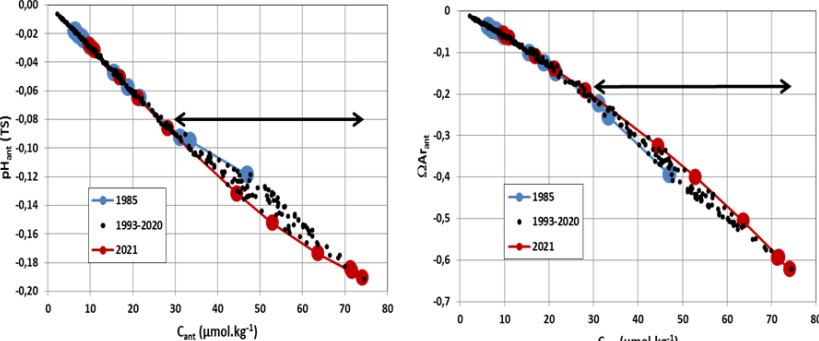

Figure 11: Anthropogenic pH ($pH_{ant}$) and anthropogenic Ωar ($Ωar_{ant}$) versus anthropogenic $CO_2$ concentrations ($C_{ant}$, µmol.kg$^{-1}$) at station OISO-KERFIX (50°40'S-68°25'E). The data are selected in the layer 150-1600m for the periods 1985 (blue), 1993-2020 (black) and 2021 (red). The arrow identifies the data in the layer 150-500m (for $C_{ant}$ > 30 µmol.kg$^{-1}$). Below 500m no change of $C_{ant}$ was observed from 1985 to 2021 and thus for $pH_{ant}$ and $Ωar_{ant}$.

The increase in $C_{ant}$ concentrations over time (Figure 6b) also leads to a decrease of carbonate ion concentrations $[CO_3^{2-}]$ and of Ωar and Ωca (Figure S14, S15). These decreases are well identified since the pre-industrial era in the whole water column but in the last 36 years, observations do not show any appreciable changes below 500m (Figure 11). The aragonite saturation state (Ωar=1) was found around 600m in 1985 and around 400m in recent years (2015-2021, Figures S14, S15). Moreover, during the period covered by observations (1985-2021), we did not detect abrupt change of the aragonite saturation horizon from one year to the next (including from season to season, Figure S16). This contrasts with previous regional studies in the SO and most notably with results from the layers close to the deep minimum of carbonate ion concentrations (Hauri et al., 2015; Negrete-Garcia et al., 2019). At our station the $[CO_3^{2-}]$ minimum lies around 500-600m (Figure S14, S15) and, along with the superimposed $C_{ant}$ accumulation, explains the upward shift of the aragonite and calcite saturation between the pre-industrial and modern periods (Figure S15). At pre-industrial time under-saturation with regard to aragonite (Ωar<1) was found at the bottom only (1600m) whereas in 1985-2021 it was found in the water column below 600 m or 400 m (Figure S15). The subsurface pre-industrial Ωar value was around 1.9-2 (Figure S15) and in the range of Ωar value in the Southern Ocean at pre-industrial time from ESM models (Jiang et al., 2023, their Figure 4).





The aragonite under-saturation already occurred in 1985 at 500-600m (Figure S15) and a small increase
of $C_T$ (via $C_{ant}$ accumulation) close to the $[CO_3^{2-}]$ minimum would rapidly shift the aragonite saturation horizon
in layers above 500m. This might have already occurred and explains that $\Omega$ar value was 1.02 at 350m in 2021
(Figure S15). These results suggest that for pelagic calcifiers living in subsurface (150m or deeper) such as
pteropods and/or foraminifera (e.g. Hunt et al., 2008; Meilland et al., 2018) the impact of acidification might
occur sooner than in surface.
For the interpretation of the trend analysis based on observations, only data below 150m could be used
as $C_{ant}$ was not evaluated in the surface layer. At 200m, based on $A_T$-$C_T$ data, pH and $\Omega$ar decreased from 1985
to 2021 by -0.059 for pH (Figure 10b) and -0.16 for $\Omega$ar (Figure S15). In 36 years, this represents about 30% of
the total change since the pre-industrial era (-0.184 for pH and -0.6 for $\Omega$ar at 200m). This is mainly linked to the
$C_{ant}$ change that represents also 30% increase in 36 years (+24.6 µmol.kg$^{-1}$ from 1985 to 2021 for a total of +71.7
µmol.kg$^{-1}$ $CO_2$ accumulated at 200m in 2021, Figure 7). We conclude that anthropogenic $CO_2$ drives the change
of the carbonate system in subsurface and probably also in surface waters.
In order to quantify the propagation of surface trends to depth, the temporal variations of carbonate
properties in the surface for both summer and winter derived from the FFNN model are compared to the changes
observed across the water column (Figure 12). The comparison shows that the seasonal amplitude of surface
waters properties was of a similar magnitude to the observed changes in the mixed layer between 1985 and 2021.
For example, the $C_T$ and $\Omega$ar seasonality, respectively around 20 µmol.kg$^{-1}$ and 0.2, corresponds to the $C_T$
increase and $\Omega$ar decrease from 1985 to 2021. The comparisons also highlight that in summer the FFNN results
were close to observations in the mixed-layer (e.g. $C_T$ was 2120 µmol.kg$^{-1}$ in 1985 and 2140 µmol.kg$^{-1}$ in 2021).
In winter the surface properties are different ($C_T$ was higher, and pH, $[CO_3^{2-}]$, $\Omega$ar were lower) and intercept the
observations at depth close to the winter water (150-200m). This is true in 1985 and 2020/2021. Specifically,
surface $C_T$ from the FFNN model in winter 1985 (2145.5 µmol.kg$^{-1}$) equaled the $C_T$ measured at 150 m in
March 1985 (2148 µmol.kg$^{-1}$). In 2020, the winter $C_T$ at the surface (2168.3 µmol.kg$^{-1}$) is equal to $C_T$
concentrations observed at 150-180 m in January 2020 or in 2021. For $\Omega$ar, the surface value derived from the
FFNN model in winter 1985 (1.6) equal to the $\Omega$ar observed at 125 m in March 1985. In 2020, the surface winter
estimate of $\Omega$ar (1.42) was equal to $\Omega$ar observed at 100-150 m in January 2020 or 2021. The same
correspondences between winter surface and WW data were identified for pH and $[CO_3^{2-}]$ (Figure 12). This
supports the use of winter and summer surface data from the FFNN model to investigate the seasonal $\Omega$ar trends
and their projection in the future.
The surface water $\Omega$ar ($\Omega$ca) trend from the FFNN model in summer of -0.0059.yr$^{-1}$ (-0.0094.yr$^{-1}$) was
stronger than in winter -0.0050.yr$^{-1}$ (-0.0079.yr$^{-1}$) and also higher than derived from observations in the WW (-
0.0043.yr$^{-1}$ for $\Omega$ar and -0.0069.yr$^{-1}$ for $\Omega$ca). The results indicate that the change of carbonate properties in the
years 1985-2021 were mainly driven by $C_{ant}$ accumulation in surface waters and across the water column.
However, potential changes in primary productivity after 2010 mitigated the effects of increasing $C_{ant}$
accumulation in response to increasing atmospheric $CO_2$ leading to relatively stable summer $C_T$ and $fCO_2$ and to
a stronger $CO_2$ sink (Figure 3). Consequently, when restricted to the period 2010-2020, the trend of $\Omega$ar in
surface waters in summer was much smaller, -0.024.decade$^{-1}$ (± 0.027) than during the preceding period. This
was much smaller than derived from the all data in 1985-2021 (-0.048.decade$^{-1}$) or estimated from reconstructed
fields in the SO-SPSS in 1982-2021 (-0.0616.decade$^{-1}$, Ma et al., 2023). It underscores the uncertainty in
extrapolating long-term time-series depending on the selection of data and periods.



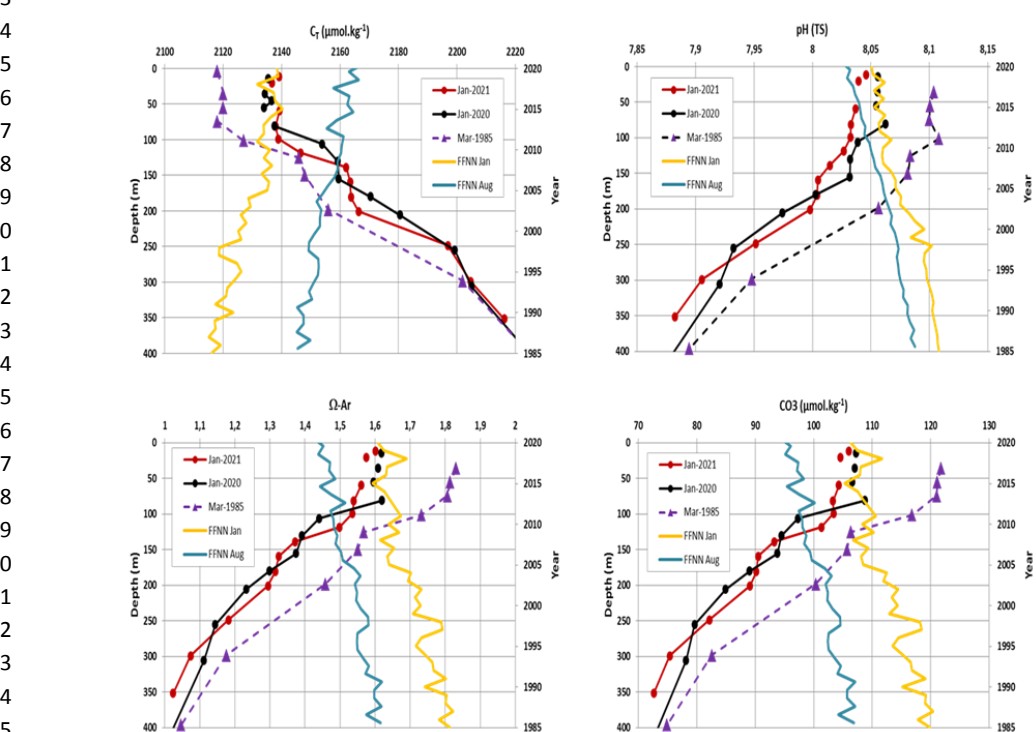

Figure 12: Profiles (0-400m left axis) of observed and calculated properties ($C_T$, pH, Ω-ar, [$CO_3^{2-}$]) at station OISO-KERFIX (50°40'S-68°25'E) in Mach 1985, January 2020 and January 2021 along with surface time-series in 1985-2020 (right axis) of the same properties in January (yellow line) and August (blue line) from the FFNN model. The FFNN values in January 2020 are coherent with January 2020 observations in the mixed-layer and in January 1985 are close to the observations in March 1985. Note that the differences of properties between 2020-21 and 1985 have a similar magnitude as the seasonal amplitude (illustrated by the FFNN values for January and August).

## 3.4 Long-term change in surface water, from the sixties to the future.

The data described above allowed evaluating the temporal variations of the properties of the carbonate system and $C_{ant}$ over 1985-2021 along with a comparison to the pre-industrial state in the water column excluding the surface layer. The results over 36 years informed on the recent changes, inter-annual variations and trends, but the time-series appears somehow short to extrapolate the trends over time. What was the change of the carbonate system in surface water before 1985 and what will be its future evolution ?

### 3.4.1 Back to the sixties: observed trends since 1962.

To explore the long-term change, we start by comparing our recent data with the observations from the LUSIAD cruise conducted in 1962-1963 (Keeling and Waterman, 1968). Some data from this cruise were obtained mid-November 1962 south of the Polar Front, in the region south-west off Kerguelen. Because of the seasonality, we compared the November 1962 data with our observations obtained in October-November in



1995, 2011 and 2016, and with the FFNN model results for November (Figure 13). The $C_T$ concentration, pH,

$\Omega$ar and $\Omega$ca for 1962 were calculated using $fCO_2$ data and $A_T$ (from the $A_T/S$ relationship Eq. 1).

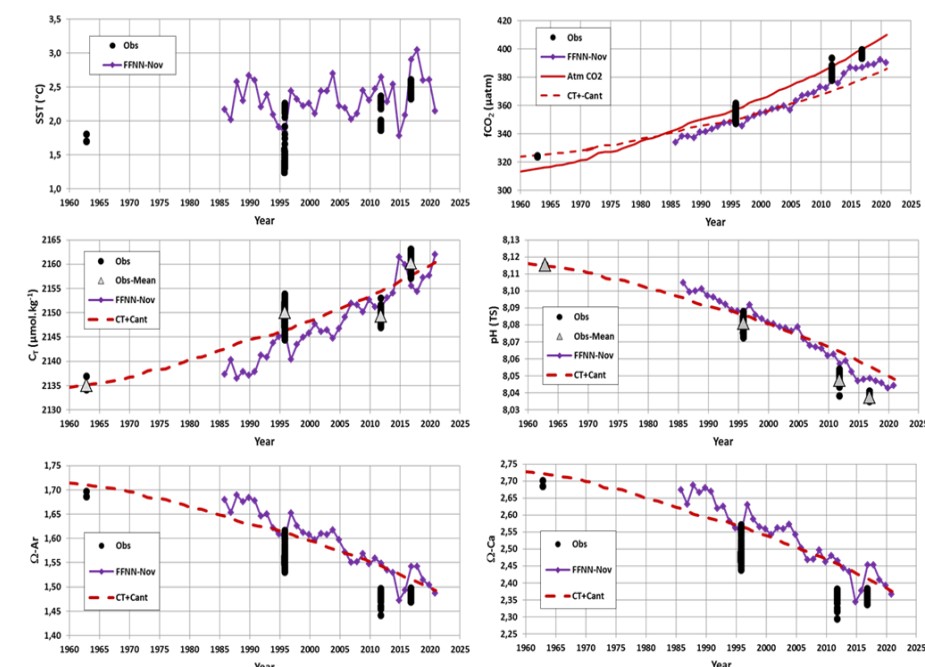

Figure 13: Observed (black dots) sea surface temperature (°C), $fCO_2$ (µatm), $C_T$ (µmol.kg$^{-1}$), pH (TS), $\Omega$-ar and $\Omega$-ca around station OISO-KERFIX at 50°40'S-68°25'E for October-November. Also shown are the results for the FFNN model for November in 1985-2020 (Purple). The $C_T$ concentrations, pH, $\Omega$-ar and $\Omega$-ca were calculated from $fCO_2$ data using the $A_T/S$ relation (Eq. 1). The red line is the atmospheric $fCO_2$ and red dashed-lines in each plot are the evolution of properties since 1960 corrected to $C_{ant}$ where $fCO_2$, pH, $\Omega$-ar and $\Omega$-ca were recalculated using $C_T+C_{ant}$, $A_T$ constant at 2290 µmol.kg$^{-1}$ and SST at 2°C. Grey triangles identified the mean values for $C_T$ and pH.

First, we note that measured SST in November 1962 (1.7°C) was slightly lower (on average about -
0.6°C) compared to recent years, but SST as low as 1.8°C for this season were also recorded in other periods
(e.g. November 1995, 2014). The change in SST is unlikely to explain the long-term increase in $fCO_2$ or
decrease in pH since 1962 (Figure 13). In 1962, the ocean $fCO_2$ was 324 µatm slightly higher than in the
atmosphere ($\Delta fCO_2$=+8 µatm, a small source), whereas in November 1985-2020 the ocean was a small $CO_2$ sink
on average ($\Delta fCO_2$= -3.3 ± 4.5 µatm). The $C_T$ concentration in 1962 (2135 µmol.kg$^{-1}$) was much lower than
observed in the 90s and the pH (8.115) much higher than in recent years (Figure 13). Compared to 1962, pH in
2016 was -0.078 lower, i.e. representing 70% of the pH decrease of -0.11 in the global ocean since the beginning
of the industrial era (Jiang et al, 2019). In November 1962, surface $C_T$ was lower by -15.1 µmol.kg$^{-1}$ compared to
the data in October 1995, i.e. a trend of +0.46 µmol.kg$^{-1}$.yr$^{-1}$ in 33 years close to the $C_{ant}$ trend observed in the
WW in 1985-2021 as described above (+0.53 ± 0.01 µmol.kg$^{-1}$.yr$^{-1}$). Having the $C_T$ value in 1962, we can
project the $C_T$ in time by adding the $C_{ant}$ concentration based on the relationship observed between $C_{ant}$ and
atmospheric $CO_2$ (Figure 7b) assuming that the anthropogenic $CO_2$ uptake since the sixties is representative of
the $C_T$ change (i.e. the change of $C_T$ due to natural variability is small). This projection is shown for all
properties (red dashed-lines in Figure 13) and confirms that the progressive $C_{ant}$ accumulation explained most of



the $C_T$ and $fCO_2$ increase in surface since 1962. We note that the $C_T$ derived from the FFNN model suggests slightly lower $C_T$ compared to the $C_{ant}$ projection especially before 2004. The difference of projected $C_T$ and the FFNN model (on average $-2.2 \pm 2.7$ µmol.kg$^{-1}$) is within the uncertainty of $C_T$ calculations (error is $\pm 5$ µmol.kg$^{-1}$ when using the $A_T/fCO_2$ pairs) and the trend of the difference over 1985-2020 ($-0.15$ µmol.kg$^{-1}$.yr$^{-1}$) is too small to be related with confidence to changes associated with natural processes. On the other hand, the ocean $fCO_2$ recalculated with the projected $C_{ant}$ trend suggested that for this season (November) the ocean moved from a $CO_2$ source in 1962-1985 ($\Delta fCO_2 > 0$) to a sink in 1986-2021 ($\Delta fCO_2 < 0$) in line with results from the FFNN model. The recalculated $fCO_2$ with $C_{ant}$ (dashed red line in Figure 13) was close to that observed in 1995 or from the FFNN model in 1985-2014 (mean difference over 1985-2014 is $-1.2 \pm 5.2$ µatm). After 2016, the recalculated $fCO_2$ suggest a stronger sink and the difference with observations in 2011 and 2016 or the FFNN model is slightly higher (mean difference over 2016-2020 is $-8.8 \pm 1.5$ µatm). Although the differences are in the range of the error in $fCO_2$ calculation using $A_T$-$C_T$ pairs ($\pm 13$ µatm), this might indicate that after 2016 a process could contribute to increase $fCO_2$ faster than the effect of $C_{ant}$ only. This difference could be due to the warming that occurred after 2016 when SST was higher than 2°C and up to 3°C in November 2017 (Figure 13 and Figure S9). The same could be applied for pH that was slightly lower than the pH recalculated from $C_{ant}$ trend after 2015 (the mean difference between recalculated pH and FFNN-pH over 1985-2020 is only $0.002 \pm 0.006$). Therefore, we conclude that for November the pH decrease since 1962 was mainly driven by anthropogenic $CO_2$. Aragonite and calcite saturation states also show a clear decrease since 1962 (Figure 13), a diminution of 11% in 59 years for both $\Omega$ar and $\Omega$ca. Based on these results over almost 60 years that confirm the conclusions from the observations in 1985-2021, we now evaluate the long-term change of the carbonate system in surface water in the future.

### 3.4.2 Projecting the observed trends in the future

The trends of the properties based on observations in 1962-2021 and the FFNN model in 1985-2020 indicate relatively linear trends linked to $C_{ant}$ uptake albeit with some decadal variability in summer (Figure 4). A simple linear extrapolation of the trends in the future suggests that aragonite saturation in surface water would be reached in year 2110 for the winter season and 2120 for summer (Figure S17) whereas the trend in subsurface suggests under-saturation in 2090. In year 2100, surface pH and [H$^+$] would be around 7.9 and 12 nmol.kg$^{-1}$ (Figure S17). However, ESM CMIP6 models suggest that under a high emission scenario (SSP5-8.5), pH in 2100 in the Southern Ocean near 50°S would be around 7.65 and [H$^+$] around 22 nmol.kg$^{-1}$ (Jiang et al., 2023, their figure 4). This suggests that the simple linear extrapolation based on recent observed trends (Figure S17) underestimated the future change of the carbonate system for a high emission scenario as previously shown in the South-Eastern Indian Ocean based on summer trends derived from observations in 1969-2003 (Midorikawa et al., 2012, their figure 4).

To better investigate the changes in the next decades, we assumed that the $C_{ant}$ trend for the modern period (Figure 7) that experienced a "business as usual" scenario after the sixties is representative of the future changes in the surface ocean carbonate system. For this analysis, we use two emissions scenarios (Shared Socioeconomic Pathways, SSP, Meinshausen et al., 2020) with atmospheric $xCO_2$ reaching 1135 ppm in 2100 (a "high" emission scenario SSP5-8.5) or $xCO_2$ reaching 603 ppm in 2100 after a stabilization around 2080 (scenario SSP2-4.5). This enables to simulate future $C_T$ concentrations for summer or winter (Figure 14) and to



calculate other carbonate properties using $C_T$ and $A_T$ (Figure 15, Table 2) in response to approximated future
changes in physical and geochemical properties excluding impacts of changes in atmospheric and oceanic
circulation. As the calculated properties are sensitive to $A_T$ values, we used a fixed $A_T$ of 2280 µmol.kg$^{-1}$ or
applied a correction based on the long-term change of sea surface salinity observed in the last 6 decades (1960-
2017), i.e. a freshening in the Southern Ocean of around -0.01 to -0.02.decade$^{-1}$ (Durack and Wijffels, 2010;
Cheng et al., 2020b). The decrease in salinity in the South Indian Ocean (-0.02.decade$^{-1}$ ± 0.01) was recently
analyzed by Akhoudas et al. (2023) who showed that in the years 1993-2021 the freshening was mainly due to
an increase in the precipitation linked to the acceleration of the atmospheric hydrological cycle. From our data in
the mixed-layer over 1985-2021, we estimated a trend in salinity of -0.0207.decade$^{-1}$ (± 0.0041). For the $A_T$
sensitivity test we thus select a salinity trend of -0.01.decade$^{-1}$ in 1962-1985 and -0.02.decade$^{-1}$ after 1985 and
apply these trends to simulate $A_T$ over 1960-2100 using the $A_T$/Salinity relationship (Equation 1). This leads to a
salinity of 33.650 and $A_T$ of 2272 µmol.kg$^{-1}$ in year 2100, about 8 µmol.kg$^{-1}$ lower than observed in 2021 (2280
µmol.kg$^{-1}$). Compared to the $C_T$ change from 2021 to 2100 (+50 and +193 µmol.kg$^{-1}$ for the "low" and "high"
emissions scenario, Figure 14), the impact of $A_T$ decrease has a minor effect on the future change for pH, $[CO_3^{2-}]$
or Ω (Table 2). For example, in winter for the SSP5-8.5 scenario, when the $A_T$ decrease is taken into account, pH
in 2100 is 7.316 and $\Omega_{Ar}$ is 0.33 against 7.372 and 0.34 when $A_T$ is constant (Table 2). In both cases, the
aragonite saturation ($\Omega_{Ar}$=1) in winter occurred in 2055, whereas in summer it is identified in 2070. The effect of
lower $A_T$ in the future appeared also small compared to the seasonal differences of pH and Ω in 2100.
As noted above, the Southern Ocean experienced a warming in recent decades (e.g. Auger et al., 2021)
and it is projected that warming will continue in the future (IPCC, 2022). Therefore, to test the sensitivity of
calculated properties to warming we applied a correction of +0.0125°C.yr$^{-1}$ in 1985-2020 and +0.025°C.yr$^{-1}$ after
2020 (Azarian et al, 2023). As for $A_T$, these results are compared for winter using constant SST (Table 2). The
effect of the long-term warming does mainly impact the projection of $[H^+]$ and pH (Table 2).
These sensitivity tests for temperature and $A_T$ showed that as for the observed period 1962-2021 (Figure
13), the projection in the future depends mainly on the anthropogenic $CO_2$ accumulation. Here, the $C_T$
concentrations were calculated using the $C_{ant}$ versus atmospheric $CO_2$ relationship (Figure 7b). We thus tested
the results for winter based on the error associated with this relationship (Figure S18). This leads to either higher
or lower $C_T$ compared to original calculation (Figure 14). For the SSP5-8.5 scenario, the winter $C_T$
concentrations in 2100 range between 2328 and 2378 µmol.kg$^{-1}$, higher than simulated in the ESM CMIP6
models around 50°S (2300 µmol.kg$^{-1}$, Jiang et al., 2023). As in the ESM models for the SSP2-4.5 scenario, the
projected $C_T$ concentration in 2100 at our location is much lower 2217 µmol.kg$^{-1}$ (Figure 14). The future change
of the carbonate system is not significantly different using low or high $C_{ant}$ accumulation (Figure S18) but this
test gives a range of years to reach aragonite and calcite under-saturation. In winter (SSP5-8.5 scenario),
aragonite would reach under-saturation between year 2050 and 2060 and between year 2070 and 2080 for
calcite. Note that for summer we derived under-saturation for $\Omega_{Ar}$ in year 2065 and for $\Omega_{Ca}$ in year 2085. For the
SSP2-4.5 scenario, where $C_T$ is 143 µmol.kg$^{-1}$ lower in 2100 compared to SSP5-8.5, aragonite under-saturation
would not be reached before 2070 (Figure 15).



Table 2: Results of the simulated properties for year 2020, 2050 and 2100 for two emission scenario (SSP5-8.5 or SSP2-4.5). For 2020 the results based on observations in January (Obs) and the FFNN model in January and August also listed. Sensitivity tests: "SSP85 W-T" is for winter with constant temperature and "SSP85 W-A-T" is for winter with constant $A_T$ and temperature.

| Method | Year | Atm-CO$_2$ ppm | fCO$_2$ µatm | $C_T$ µmol.kg$^{-1}$ | $A_T$ | pH TS | [H$^+$] nmol.kg$^{-1}$ | [CO$_3^{2-}$] µmol.kg$^{-1}$ | Ωca | Ωar |
|---|---|---|---|---|---|---|---|---|---|---|
| Obs Jan | 2020 | 410.6 | 391.9 | 2142.2 | 2281.8 | 8.044 | 9.04 | 105.2 | 2.53 | 1.59 |
| Std obs. | | | (2.0) | (0.7) | (0.3) | (0.002) | (0.04) | (0.5) | (0.01) | (0.01) |
| FFNN Jan | 2020 | 410.6 | 385.1 | 2138.5 | 2280.1 | 8.051 | 8.90 | 106.3 | 2.55 | 1.61 |
| SSP Summer | 2020 | 414.9 | 375.4 | 2137.5 | 2282.1 | 8.061 | 8.70 | 108.0 | 2.60 | 1.63 |
| FFNN Aug | 2020 | 410.6 | 410.0 | 2168.3 | 2289.8 | 8.024 | 9.45 | 94.2 | 2.27 | 1.42 |
| SSP Winter | 2020 | 414.9 | 434.5 | 2167.3 | 2282.1 | 8.001 | 9.98 | 90.4 | 2.18 | 1.37 |
| | | | | | | | | | | |
| SSP85 Summer | 2050 | 562.8 | 526.5 | 2177.2 | 2278.3 | 7.928 | 11.79 | 84.2 | 2.02 | 1.28 |
| SSP85 Winter | 2050 | 562.8 | 624.7 | 2207.0 | 2278.3 | 7.857 | 13.91 | 68.5 | 1.65 | 1.04 |
| SSP85 W-A-T | 2050 | 562.8 | 585.7 | 2207.0 | 2280.0 | 7.880 | 13.17 | 69.0 | 1.66 | 1.04 |
| SSP85 W-T | 2050 | 562.8 | 592.7 | 2207.0 | 2278.3 | 7.875 | 13.32 | 68.1 | 1.64 | 1.03 |
| SSP45 Winter | 2050 | 506.9 | 554.8 | 2192.0 | 2278.3 | 7.905 | 12.46 | 75.8 | 1.92 | 1.15 |
| | | | | | | | | | | |
| SSP85 Summer | 2100 | 1135.2 | 1986.9 | 2330.6 | 2271.8 | 7.394 | 41.31 | 26.9 | 0.65 | 0.41 |
| SSP85 Winter | 2100 | 1135.2 | 2306.3 | 2360.4 | 2271.8 | 7.316 | 48.26 | 21.8 | 0.52 | 0.33 |
| SSP85 W-A-T | 2100 | 1135.2 | 1993.1 | 2360.4 | 2280.0 | 7.372 | 42.44 | 22.6 | 0.54 | 0.34 |
| SSP85 W-T | 2100 | 1135.2 | 2097.0 | 2360.4 | 2271.8 | 7.349 | 44.74 | 21.3 | 0.51 | 0.32 |
| SSP45 Winter | 2100 | 602.8 | 753.9 | 2217.7 | 2271.8 | 7.782 | 16.51 | 60.9 | 1.47 | 0.92 |

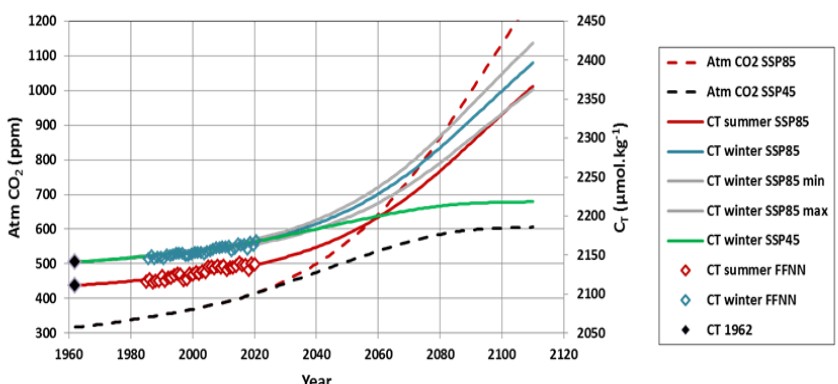

Figure 14: Evolution of atmospheric CO$_2$ (ppm) and sea surface $C_T$ (µmol.kg$^{-1}$) in 1960-2110 evaluated for 2 scenarios (SSP2-4.5 black dashed and SSP5-8.5 red dashed), for summer (red line for SSP5-8.5) and winter (blue line for SSP5-8.5 and green line for SSP2-4.5). Grey lines are the high and low $C_T$ for winter SSP5-8.5 based on the error in the $C_{ant}$/fCO$_2$ relationship (figure 7b). Also shown are the results for the FFNN model in 1985-2020 for summer (red diamonds) and winter (blue diamonds) and $C_T$ in 1962 (black diamonds). The $C_T$ values for different seasons and scenarios were used to calculate the carbonate properties in the future (Figure 15).



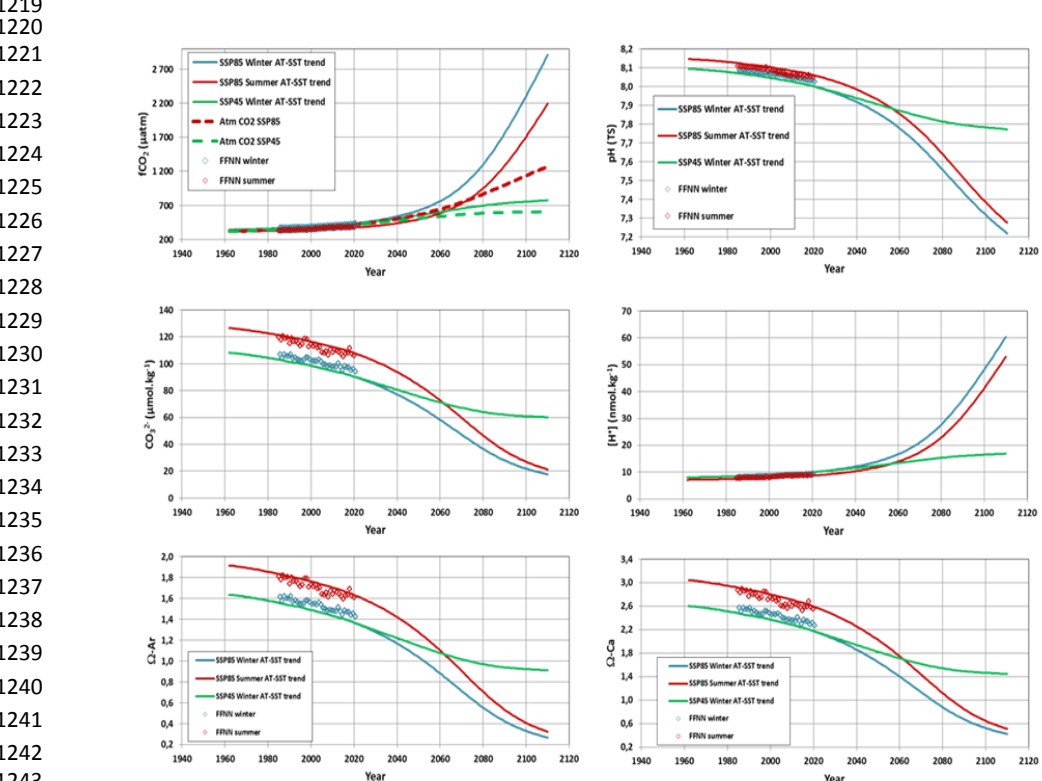

Figure 15: Evolution of sea surface $fCO_2$ (µatm), pH (TS), $[CO_3^{2-}]$(µmol.kg$^{-1}$), $[H^+]$ (nmol.kg$^{-1}$), Ω-Ar and Ω-Ca in 1960-2110 evaluated for the SSP5-8.5 scenario for winter (blue line) and summer (red line) taking into account both $A_T$ and SST future trends. For winter results are also presented using the SSP2-4.5 scenario (green line). Also shown are the results for the FFNN model in 1985-2020 for summer (red diamonds) and winter (blue diamonds). Atmospheric $fCO_2$ is also shown for SSP5-8.5 (red dashed) and SSP2-4.5 (green dashed). Values in 2020, 2050 and 2100 for different sensitivity tests are listed in Table 2.

## 4 Summary and concluding remarks

The times-series of high quality observations in 1985-2021 and the results from the FFNN model at one location, south of the Polar Front in the Southern Indian Ocean (50°S-68°E) presented in this analysis offered new results on the inter-annual variability, decadal to long-term trends of the carbonate system in surface waters, air-sea $CO_2$ fluxes and associated drivers. The evaluation of anthropogenic $CO_2$ concentrations in the water column indicates that the trends of the carbonate species are mainly driven by the $CO_2$ uptake leading to a progressive acidification in surface waters and at depth.

In 1985, the $C_{ant}$ concentrations were approaching 50 µmol.kg$^{-1}$ at 200 m and $C_{ant}$ was detected in the water column down to the bottom (1600m). This explains why aragonite under-saturation was observed at around 600m in 1985, where $[CO_3^{2-}]$ concentration was at minimum, whereas at pre-industrial era the water column was super-saturated (this study Figure S15; Lauvset et al., 2020, their Figure S15). 36 years later because of the anthropogenic $CO_2$ accumulation, we observed an upward migration of the aragonite saturation horizon that was found around 400 m in 2021 (a shoaling rate of around -6 m.yr$^{-1}$).



At subsurface, in the winter water layer, the $C_{ant}$ trend is estimated at +0.53 (± 0.01) µmol.kg$^{-1}$.yr$^{-1}$ in

1985-2021 with a detectable increase in recent years (up to 72 µmol.kg$^{-1}$ in 2021 compared to 47 µmol.kg$^{-1}$ in
1985). The $C_{ant}$ concentrations in the ocean are closely related to the atmospheric $CO_2$ concentrations and the
slope we observed south of the PF in the Indian sector of +0.263 ± 0.042 µmol.kg$^{-1}$.µatm$^{-1}$ is close to that
observed in the AAIW in the South Atlantic (+0.23 ± 0.05 µmol.kg$^{-1}$.µatm$^{-1}$, Fontela et al., 2021). This suggests
that local observations in the South Indian POOZ captured the link between $C_{ant}$ and atmospheric $CO_2$ at larger
scale.

In surface waters, over 1991-2020 the oceanic $fCO_2$ increased at a rate close or slightly lower than in

the atmosphere (Figure 2b) and the $C_T$ trend followed the $C_{ant}$ accumulation (Figure 4b, S12a). However in the
last decade both observations and the FFNN model showed low $fCO_2$ trends in summer (less than 1 µatm.yr$^{-1}$).
The change in summer trend appears related to primary production as revealed by a decrease of Chl-a in 1998-
2010 followed by an increase after 2010. Biological activity counteracts the $C_T$ increase due to $C_{ant}$, resulting in
rather stable $C_T$ and $fCO_2$ in summer 2010-2020 (+0.38± 0.26 µmol.kg$^{-1}$.yr$^{-1}$ and +0.98 ± 0.40 µatm.yr$^{-1}$). As a
result, the region moved from an annual source of +0.8 molC.m$^{-2}$.yr$^{-1}$ in 1985 to a sink of -0.5 molC.m$^{-2}$.yr$^{-1}$ in
2020. The increase of the ocean $CO_2$ sink was particularly pronounced after 2011 (Figure 3) when phytoplankton
biomass was stronger in this HNLC region and occurred when the SAM index was in a positive state.

In 1959-1963, the SAM was also positive on average and moved to a negative phase in 1964 (Marshall,

2003; King et al., 2023). Historical data from 1962 suggest that in November the region was a small $CO_2$ source
($\Delta fCO_2$=+8 µatm). Assuming the seasonality was the same as in the 80s, we estimate that in 1962 the annual flux
would be around 2.2 molC.m$^{-2}$.yr$^{-1}$. Extrapolating to the entire South Indian POOZ (50-58°S/20-120°E, 6.5
Mkm$^2$), this region was a $CO_2$ source of 0.17 PgC.yr$^{-1}$ in 1962, reduced to 0.06 PgC.yr$^{-1}$ in 1985 and a $CO_2$ sink
of -0.04 PgC.yr$^{-1}$ in 2020. This could be compared with reconstructed fluxes from a data-based model that
produced a $CO_2$ source in 1960-1990 and a sink in 2020 in the south Indian sector (Rödenbeck et al., 2022, their
Figure 6).

For November 1962, the estimated $C_T$ concentration in surface (2135 µmol.kg$^{-1}$) is 21 µmol.kg$^{-1}$ lower

than observed mid-October 2016 in the mixed-layer (2156 µmol.kg$^{-1}$). This is almost equal to the increase of $C_{ant}$
in 54 years (+22.3 µmol.kg$^{-1}$). As a result, surface ocean pH dropped from 8.11 in 1962 to 8.044 in 2020. Over
multi-decadal time scale (30 years or more), acidification in the South Indian POOZ has been mainly controlled
by uptake of anthropogenic $CO_2$. However, our data also indicate a modulation of the summer pH trend by
natural processes. After 2010, a very small pH trend was estimated in summer (-0.0098.decade$^{-1}$ ± 0.0042) when
the region experienced higher primary productivity. On the opposite, in winter, the pH trends continuously
increased with time, -0.010.decade$^{-1}$ (± 0.001) in 1991-2001 and -0.021.decade$^{-1}$ (± 0.002) in 2010-2020. In the
subsurface (winter water layer), the trend of pH based on $A_T$-$C_T$ data in 1985-2021 of -0.0161 (± 0.0033).decade$^{-}$
$^{1}$ is also almost equal to the annual surface trend from the FFNN model. A simple extrapolation of the trends in
the WW indicated that under-saturation ($\Omega$<1) would be reached at year 2090 for aragonite and year 2180 for
calcite. However, as atmospheric $CO_2$ will desperately continue to rise and ocean $C_T$ will increase in the future,
the pH and $\Omega$ will decrease at a faster rate than observed in recent years. A projection of future $C_T$ concentrations
based on emissions scenario, excluding changes in ocean circulation, indicated that the winter surface pH in
2100 would decrease to 7.32 for a high emission scenario (SSP5-8.5) or to 7.782 for a low emission scenario
(SSP2-4.5). This is up to -0.86 lower than pre-industrial pH and -0.71 lower than pH observed in 2020. For the



winter season the aragonite saturation in surface would be reached around 2050 for a high emissions scenario
and 2070 for a low emission scenario.

The time-series presented here for the Southern Ocean, along with other historical time-series of $A_T$-$C_T$
in the water-column (BATS, HOT, ESTOC, KNOT, Iceland or Irminger seas; Bates et al., 2014; Lange et al.,
2023) or the recent BG-Argo floats in the Southern Ocean (Mazloff et al., 2023) offer useful data for the
evaluation of biogeochemical and Earth system models, especially the coupling of $fCO_2$, $C_T$, $A_T$, and pH not well
represented in current models at seasonal to decadal scale in the Southern Ocean (e.g. Hauck et al., 2023;
Rodgers et al., 2023; Joos et al., 2023). Observing the decadal changes of the carbonate system in the water
column is also an important step to extend the evaluation of biogeochemical and ESM models below the surface
(Jiang et al., 2023). It is important to maintain such time-series for monitoring the future evolution of the ocean
$CO_2$ sink, of the acidification and its impact on phytoplankton species and higher trophic levels. This is
especially the case in Marine Protected Area such as the French Sub-Antarctic islands including the Kerguelen
Archipelago which was listed as a UNESCO World Heritage site in 2019.
**Data availability:**
Data used in this study are available in SOCAT (www.socat.info) for $fCO_2$ surface data, in GLODAP
(www.glodap.info) for water-column data and at NCEI/OCADS (www.ncei.noaa.gov/access/ocean-carbon-data-
system/oceans/VOS_Program/OISO.html). The CMEMS-LSCE-FFNN model data are available at E.U.
Copernicus Marine Service Information (https://resources.marine.copernicus.eu/products).
**Authors contributions:**
CLM and NM are co-I of the ongoing OISO project. CLM, NM, CL and CR participated to OISO cruises.
Underway $fCO_2$ was measured by CLM, NM, CL, and qualified by CLM and NM. Nutrients data were measured
and qualified by CLM and CL. Chl-a data were measured and qualified by CR. Water column data were
qualified by CLM, NM, CL, CR and GR. MG, FC and TTTC developed the CMEMS-LSCE-FFNN model and
provided the model results. NM started the analysis, wrote the draft of the manuscript and prepared the figures
All authors contributed to revising the draft manuscript.
**Competing interest:** The authors declare that they have no conflict of interest.

**Acknowledgments**: The OISO program was supported by the French institutes INSU (Institut National des
Sciences de l'Univers) and IPEV (Institut Polaire Paul-Emile Victor), OSU Ecce-Terra (at Sorbonne Université),
and the French program SOERE/Great-Gases. We thank the French oceanographic fleet ("Flotte
océanographique française") for financial and logistic support for the OISO program
(https://campagnes.flotteoceanographique.fr/series/228/). We thank the captains and crew of *R.R.V. Marion
Dufresne* and the staff at IFREMER, GENAVIR and IPEV. We also thank Jonathan Fin and Claude Mignon for
their help during the OISO cruises. The development of the neural network model benefited from funding by the
French INSU-GMMC project "PPR-Green-Grog (grant no 5-DS-PPR-GGREOG), the EU H2020 project
AtlantOS (grant no 633211), as well as through the Copernicus Marine Environment Monitoring Service (project
83-CMEMS-TAC-MOB). We thank all colleagues that contributed to the quality control of ocean data made
available through CARINA and GLODAP (www.glodap.info). The Surface Ocean $CO_2$ Atlas (SOCAT,



www.socat.info) is an international effort, endorsed by the International Ocean Carbon Coordination Project
(IOCCP), the Surface Ocean Lower Atmosphere Study (SOLAS) and the Integrated Marine Biogeochemistry
and Ecosystem Research program (IMBER), to deliver a uniformly quality-controlled surface ocean CO2
database.

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
