# Peer review of "Anthropogenic CO2, air-sea CO2 fluxes and acidification in the Southern Ocean: results from a time-series analysis at station OISO-KERFIX (51°S 68°E)."

_EGUsphere, 2023_

## Referee Comment (RC2)

Review of Metzl et al: Anthropogenic CO2, air-sea CO2 fluxes and acidification in the Southern Ocean.

The paper describes a comprehensive data set of in situ observations spanning multiple decades that is backed up by a gap-filled surface ocean pCO2-product in the Indian sector of the Southern Ocean. It provides critical numbers for anthropogenic CO2 uptake and accumulation, acidification rates etc. It also uses an older data set from the 1960s to put the data into perspective and uses the relationship between atmospheric CO2 and anthropogenic carbon to extrapolate the results into the future based on atmospheric CO2 of two emission scenarios. This is a very valuable scientific contribution and I fully support its publication. Thank you for collecting these data and making them available. It is particularly interesting to see the decadal variability in the carbonate system variables and how an increase in biological production can compensate the anthropogenic CO2 uptake in recent years to lead to a relatively stable carbonate system. I have a couple of comments and clarifying questions that I would like the authors to address. The list is lengthy, but many are detailed comments on sentences being not clear.

General:
1) The paper is rather long and very detailed. It is certainly a strength of the paper to provide all numbers (convincing evidence) and this is appropriate for a publication in Ocean Sciences. I would nevertheless recommend the authors to see if the text can be somewhat shortened and stream-lined, for example by moving some of the numbers into a table.
    a. In particular, I would ask the authors to shorten the summary section, which does not need to present all numbers again.
    b. Also, I was a bit confused about the sections 3.3.1 "surface pH trend" and "3.3.2 Temporal changes in the water column". The "surface" section 3.3.1 also comments on the depth profiles, which should be covered in the following section.
2) The figures are not of highest quality, probably plotted in Excel? Maybe consider whether there is a way to use a higher-quality plotting software for future work. I will list specific comments below, on font size, colors, missing legend entries where applicable. Font sizes are also generally too small in the supplementary figures.
3) Given the availability of a decadal time-series, could you comment on the discussion of the variability of the Southern Ocean carbon sink? E.g. the stagnation of the Southern Ocean carbon sink in the 1990s and/or the reinvigoration in 2015 (Le Quere et al., 2007, doi: 10.1126/science.1136188Landschützer et al., 2015, doi:10.1126/science.aab2620). You touch on SAM, but I would appreciate a clear statement on decadal variability and its drivers in the summary/concluding section.

Specific comments:
1) There are quite a few language issues, I will mention some in the technical corrections, but the list is not exhaustive. Also, quite a few times, the sentences weren't super clear. See comments below. In general, please double-check whether it is always crystal-clear whether you talk about numbers/trends from in situ data or from FFNN.

2) Line 21: "At the surface during austral winter the oceanic fCO2 increased at a rate close or slightly lower than in the atmosphere. " I'd appreciate if you added what that means for the ocean sink.

3) Line 28 "desperately". I suggest changing to "is expected to increase" or such. How atmospheric CO2 evolves depends on the decisions made by human societies.

4) Line 44 and elsewhere: why do you use parentheses around uncertainties? They belong to the number, so I would just report them in plain text (no parentheses)

5) Line 45: could update to Friedlingstein et al., 2023, 10.5194/essd-15-5301-2023

6) Line 54: "ongoing debate" add "on…" (explain the debate a little bit).

7) Lines 55-64: please mention that there is some doubt about the previously reported magnitude of decadal variability in the Southern Ocean carbon sink based on tests with the mapping products (Gloege et al., 2021, 10.1029/2020GB006788, Hauck et al., 2023, 10.1098/rsta.2022.0063)

8) Line 57 here and elsewhere you cite Hauck et al. 2023: "Sparse observations…", doi: 10.1098/rsta.2022.0063 though you probably meant to refer to the RECCAP chapter (Hauck, J.; Gregor, L.; Nissen, C.; Patara, L.; Hague, M.; Mongwe, P.; Bushinsky, S.; Doney, S. C.; Gruber, N.; Le Quéré, C.; Manizza, M.; Mazloff, M.; Monteiro, P. M. S.; Terhaar, J. The Southern Ocean Carbon Cycle 1985–2018: Mean, Seasonal Cycle, Trends, and Storage. *Global Biogeochemical Cycles* **2023**, *37* (11), e2023GB007848. https://doi.org/10.1029/2023GB007848.)

9) Line 72: upwelling → enhanced upwelling

10) ~Line 74: I would personally add a note on the suggested secondary saturation horizon that is expected as the anthropogenic carbon uptake is strongest at the surface (Hauck et al., 2010; Negrete-Garcia et al., 2019). You seem not to confirm this with your results, so might be worth introducing this here in the first place.

11) Line 90: "modest source of CO2": Combined with the next sentence it reads as if this is a model flaw. A large sink certainly is, but a modest source is possible I would say. We have very few preindustrial observations ;) and so don't know the number well.

12) Line 91-98: I would tone down the comment on the models a bit, because, yes there are systematic flaws, but also, often they fall into the observational range and/or we don't really have the necessary process understanding that would be needed to cure those uncertainties/biases.

13) Line 175: can you comment on whether the Leseurre et al paper is based on exactly the same paper?

14) Line 190: maybe add Hauck & Völker, 2015, 10.1002/2015GL063070

15) Figure 1: please increase fontsize of all text on axes and within figure. Yellow text "Kerguelen" is a bit hard to read

16) Line 289: please give a quick summary of accuracy of 1990-1995 data

17) Please comment on accuracy of 1960s data

18) Line 298: CRMs were used for all data since 1998 and for none before that?

19) Line 322: can you please add the section number where this is shown.

20) Section 2.2.3: please comment on 1960s data as well.

21) Line 359: suggest to delete the sentence on pH here, as this is the topic of the next paragraph and alkalinity has not yet been introduced.

22) Line 419 "although": add that there also wasn't much winter data for training?

23) Line 420-423: I'm confused about this sentence. Why is it here and what is the message? This is the methods section, not results.

24) Table 1 and Figure 2 provides averages over certain time-periods, but these are not defined/explained until the end of section 3.1. Please explain this earlier (methods, even if referring to results). I am also puzzled over the chosen *five* seasons, some of them being single months. Why not stick to DJF, MAM, JJA, OND? This needs justification. Also for the time-periods considered, it would be useful to also report numbers for "standard decades" 1990-1999, 2000-2009 etc even if in supplement. The chose time periods also use the "edge" years twice, e.g. 2001-2010 and 2010-2020 both have 2010 included. Is that on purpose?

25) Line 430: "sampling locations were mainly reoccupied in austral summer": unclear, where do the data for different seasons then come from? Do you simply mean: "…different seasons, though most of them stem from summer"?

26) Figure 2 and 4: please increase font size of all text (axes labels, legend). The inserted map is far too small. Colors of the symbols in the legend are not readable. Suggest changing the purple to grey or such, red symbols on top of purple line are hard to read. Still puzzled about the chosen seasons (that don't seem to be interpreted much) and decadal periods.

27) Line 487: looks to me the positive SAM started rather in 2008 than in 2010. There were also positive phases earlier.

28) Line 506: "part of": how much?

29) Figure 3: increase font sizes.

30) Figure 5: increase all font sizes. Change decimal comma "," to "." Orange dashed line missing in legend.

31) Line 590: "stability" this needs explanation before figure is shown, or delete in the caption.

32) Line 600: "albeit…" does this refer to October? Unclear.

33) Line 614-615: how does this increasing chl fit with no trend observed in nutrients? Everything used up immediately? Hauck et al., 2013 also found in a modelling study that increasing NPP can counteract the outgassing during a positive SAM phase. Would be nice to at some point in the discussion comment on the system response to SAM (summarize your findings).

34) Line 627: "higher": higher than what? It is still well below atm $CO_2$.

35) Line 637: this information is needed much earlier. I would also welcome a more detailed justification of those periods, it seems a bit arbitrary here.

36) Line 649: add depth range (100-300 m?)

37) Figure 6: Increase font sizes of axes and color bar labelsI know Cant detection methods can result in negative values, but it would be good to comment on them (artefacts or real?) What is the lowest value that you find?

38) Line 674: "top layers (0-400m, Figure 6b): Figure 6b only starts at 200 m…

39) Line 679-681: suggest ordering from surface to depth, so change order of these two sentences.

40) Line 693: "natural variability". I'm curious whether more details on processes at work can be given?

41) Line 695: explain links between CT, O2, temp and Cant here.

42) Figure 7: moderate increase in font sizes

43) Line 729: I'm curious why none of these data was used by Gruber et al? Is it not in Glodap?

44) Line 744-746: "CT trend faster in summer <-> negative CT anomaly" doesn't make sense to me, more CT or less?

45) Line 749: "twice the rate … that could be explained": how much can be explained and how was that estimated?

46) Line 753: "processes at the surface": which ones?

47) Line 756/57: I'm a bit lost which information comes from in situ obs and which from FFNN.

48) Line 759: "the temporal change" → absence of temporal change

49) Figure 8: subscripts for $CO_2$ in the legend would be nice.

50) Line 811ff: I am not sure how useful it is to list values from different sources which are all from different periods. This is acknowledged in the next sentence, but maybe this would a good place to shorten and refer to the numbers in the table without repeating them in the text.

51) Figure 9: see comments on Figure 2. I guess TS stands for total scale. Please spell out in the caption.

52) Line 863: the first number is from in situ obs? It would help me and maybe also other readers to specify.

53) Line 894: "no trend is observed for pH-PI": then why do you give a trend number in the figure?

54) Figure 11: increase font sizes. It would be easier to read if a box or background shading would be used to indicate the data for depth < 500 m (instead of the arrow). Otherwise, the arrow should be labelled directly. Also mark and label the data for depth > 500 m.

55) Line 935: "no change of Cant". Well, the figure shows between 0 and 30 µmol/kg Cant, this is not 0.

56) Line 939: "any appreciable": how defined?

57) Line 940: saturation state → saturation horizon!!

58) Line 947: saturation → saturation horizon

59) Line 960ff: the percentage numbers, do they really refer to pH or to H+. H+ would make more sense to me given that pH is a logarithmic scale.

60) Line 965: carbonate properties → carbonate system properties

61) Line 969: seasonality → seasonal amplitude

62) Line 972: I do not understand what this sentence is meant to say.

63) Line 977-981: I am confused about this part, I guess you mean to say that WW layer data can be compared to surface data, but please simplify the sentences.

64) Figure 12: increase font sizes. I found it very confusing to have the FFNN data with time on the y-axis plotted over the profiles of in situ data. After a while I understood, but it would be much appreciated if this could be simplified. An alternative could be to make a second panel for the FFNN data where color of dots is used to mark the years. This could be a smaller panel with only CT on the x-axis, no y-axis and would avoid the impression of a depth profile. Caption refers to "Jan 2020 obs", but only Jan 2021 obs are in legend.

65) Line 1039: does this calculation use salinity from 1962 or a climatology?

66) Figure 13: increase all font sizes.

67) Line 1071: 2014: I can't see SST in 2014 sticking out in Figure 13a. (2012?)

68) Line 1096: did you calculate the effect of the Delta T on pCO2?

69) Line 1110: "extrapolation of trends": do I understand correctly that this is using the trend/year and multiplying with number of years? i.e. independent of atm $CO_2$ assumptions? Please specify.

70) Line 1140/41: the aragonite saturation → the surface aragonite undersaturation

71) Line 1145: "correction": this is not a correction, but a sensitivity test. Maybe simply say "warming"? Also note that this offline approximation of temperature will overestimate the temperature effect as it neglects circulation and mixing (warming will be limited by how much not-yet-warmed water is brought into the surface mixed layer)

72) Table 2: I am a bit lost here. First: why is SSP2-4.5 (the more realistic emission scenario) only shown for winter, and not also for summer (and the test cases)? Another idea might be to list the test cases in another table so that the design of these can be grasped quicker. Short for SSP2-4.5 is usually ssp245 and for SSP5-8.5 it is ssp585 (ssp85 → ssp585). I also overlooked the ssp45 in the table entirely for a while. Could you add a white space between the lines with the high and low emission scenario?

73) Figure 14 and 15: increase font sizes, use subscripts in legend, use complete name for scenarios.

74) Line 1260: the → anthropogenic

75) Line 1264: at minimum → at its minimum.

76) Line 1269: detectable increase: of what? Concentration or trend?

77) Line 1283: interesting, and it would be nice to comment on the complete effect of positive SAM index on the $CO_2$ system. Upwelling and outgassing limited to further south? At this location mostly nutrient effects?

78) Line 1290: any change of giving some numbers for source and sink from Rödenbeck et al?

79) Line 1303: "desperately" see comment 3 above.

80) Line 1313: "coupling of .. not well represented..." : I think this is a bit too general. The coupling between CT, AT, fCO2 and pH is actually well represented with carbonate chemistry routines. It is more some of physical drivers and most of the biological drivers that cannot be represented with sufficient detail/process understanding.

Technical corrections (not complete):

Line 19: In subsurface → In the sufsurface

Line 41: grammar, "taking up a large part ... since decades"

Line 49: grammar

Line 65: is → in

Line 87: reach 0.7 → reach up to 0.7

Line 89: ESM → ESMs

Line 123: are in bracket → are given in brackets

Line 309: be not → not be

Line 321: latitude → latitudes

Line 343: excepted → except

Line 565: would occurred → would have occurred

Line 742: "count" → contribute to XX

Line 899: grammar

Line 990: the all → all the
Line 1066: corrected to → corrected for
Line 1158/59: grammar
Line 1312: BG → BGC

---

## Author Comment (AC1)

Replies to reviewer 1 (Her/his Report published 22/12/23)

Article : Metzl, N., Lo Monaco, C., Leseurre, C., Ridame, C., Reverdin, G., Chau, T. T. T., Chevallier, F., and Gehlen, M.: Anthropogenic CO2, air-sea CO2 fluxes and acidification in the Southern Ocean: results from a time-series analysis at station OISO-KERFIX (51°S-68°E), EGUsphere [preprint], https://doi.org/10.5194/egusphere-2023-2537, 2023.

The reviewer's comments and questions are written in black; our replies are in red.

;;;;;;;;;

Nicolas Metzl et al. present 36 years of measurements of carbonate system properties in the Indian Southern Ocean, an area important for oceanic CO2 uptake. They investigate multi-decadal trends and their causes. Among other results, they find the accumulation of anthropogenic CO2 to be the dominant control on carbon and acidification trends in their study area, modulated by natural decadal variability. Metzl et al. use the observed trends to extrapolate into the future, including estimates of the time of crossing undersaturation thresholds important for marine life. The findings are presented clearly and convincingly. I certainly recommend to publish this work. I would like to add that the sustained efforts to maintain high-quality oceanic measurements over such a long time frame already represent a great value for the research community in itself, far beyond what can be discussed in a single paper.

Response: We warmly thank the reviewer for her/his supportive report.

Minor comments meant to further improve clarity:

Title: consider mentioning "Kerguelen" as a more easily recognizable geographic term
Response: Thank you for the suggestion, but we prefer not to mention "Kerguelen" in the title, because the analysis focused on a region SW of the Islands (in the POOZ). These data correspond to an HNLC area, and are not in the north-east or the south-east of Kerguelen Islands (fertilized areas with sustained blooms) that are more commonly associated with the Islands.

Line 59: Rödenbeck et al (2022) note that interannual variability before the 1990s may be underestimated because there are hardly any pCO2 data to add variations not captured by the extrapolation based on relationships to predictor variables
Response: We agree with this comment. Here we wanted to refer to recent data-products that evaluate the CO2 sink back to 1957. No change.

Lines 252-253: Why would the lower salinity speak against being representative?
Response: The change in salinity might inform on different water masses (occasionally) or variation in E/P budget (for decadal and long-term trend, as recently observed, Akhoudas et al, 2023 or for future scenario as discussed in the section 3.4.2). This would potentially impact N-AT, N-CT or pH trends. Here we find that salinity surface data showed some low values in 2011-2013, but such signal has no impact on our long-term analysis (i.e. the period 2011-2013 was thus not filtered in the trend calculations). We also tested the impact of salinity change in the future and conclude (line 1138) that "the impact of AT decrease has a minor effect on the future change for pH".

Line 258: Just to clarify: Will the data not be present in any upcoming GLODAPv2.20xx release?
Response: The recent OISO stations data are not part of the current GLODAP version (Lauvset et al, 2024, ESSD submitted), but will be part of the following one.

Line 433: The seasonal cycle is actually difficult to see from Fig 2a

Response: Figure 2a aimed at presenting all data available and results from the FFNN model; this indicates that data exist for different seasons, but for trend analysis, only summer data could be used. The seasonal cycle is more clearly seen in Supp Mat Figure S4 for 1985. No change.

Line 435: Somewhat unclear: do you mean the annual $CO_2$ flux 1985-1998, or the winter one?
Response: Thank you for this comment. Yes, this is for the annual flux as shown in figure 3. The sentence will be revised as follows: "The model also indicates that between in 1985 and the mid-1990s the $fCO_2$ during austral winter (May-September) was always higher than the atmospheric $fCO_2$ leading to an annual $CO_2$ source during this period (Figure 3)."

Line 494: It sounds as if this rate has been calculated from just the difference of 2 values. If so, why not from a linear fit?
Response: The reviewer is correct, here we calculated the trend from only 2 values (the first in Feb-1991 and the last in Feb-2021): 394.9-344.4=50.5 µatm, i.e. +1.68 µatm/yr over 30 years. This very simple calculation was provided as a hint on how the trend would compare to the atmospheric $CO_2$ trend. We have changed the sentence as follows: "From the first underway measurements obtained at the OISO-KERFIX site in February 2021 to the last measurements used in this study in February 1991, the average oceanic $fCO_2$ increased by +50.5 µatm (from 344.4 ±1.5 µatm to 394.9 ±1.5 µatm, Figure 2a). During the same period, the atmospheric $CO_2$ increased by 57 µatm in this region (recorded at Crozet Island, Dlugokencky and Tans, 2022). This first comparison of two cruises 30 years apart indicates that the oceanic $fCO_2$ increase was close to that of the atmosphere."

Fig 5: I was wondering whether the break point between the 2 trends (orange) wasn't mainly induced by the large values in 1998 and 2000? Visually, the dip in 2008-2010 actually seems to be in the range of the general variability since 2002. Is is really justified to suggest a break in trends?
Response: We explored the Chl-a variability to see if one can interpret the change of the decadal $fCO_2$, CT trends due to biological processes. Remote sensing observations (Seawifs) are only available since 1998. During the first period (1998-2010), summer Chl-a appeared higher in 1998, 2000, 2001, 2005, while it did not exceed 0.2 mg/m3 over a relatively long period after that (2006-2013), and since 2008-2010 there is a progressive increase of Chl-a, with some interannual variations but lower than the large anomalies as detected in 1998 or in 2000 at that location. Other analyses at larger scale (basin scale in the SO or HNLC biomes) also suggest a change of Chl-a/productivity before and after 2010 (Basterretxea et al., 2023). We thus think that it is useful to separate the information into two periods: 1998-2010 and 2010-2021.

Lines 682-683: The change is termed "very small", but isn't it actually outside the given uncertainty range?
Response: The reviewer is correct; the difference of 2.72 µmol.kg-1 between 1985 and 2021 is slightly higher than the observed variability (±1.27 and ±0.62 µmol.kg-1 in 1985 and 2021 respectively); this is however a small difference compared to the uncertainty of the TrOCA method used to calculate Cant (±6 µmol.kg-1). We thus estimate that the derived Cant trend in the deep waters is "very small" (Figure R1, no significant decrease or increase). No change.

[Figure]

Figure R1: Cant in layers 125-150m and below 1200m. At depth, the signal of the trend is "very small".

Line 694: can you briefly indicate why lower O2 explains lower Cant?
Response: To calculate $C_{ant}$ concentrations in the water column, one separates the contribution of $C_T$ change due to organic matter remineralization. This is achieved using O2 data with an adapted Redfield ratio; following the concept of a quasi-conservative tracer such as NO (Broecker, 1974), TrOCA is a tracer based on a combination of CT, AT and O2:
TrOCA = O2 + a ($C_T$- $A_T$/2) and $C_{ant}$ = (TrOCA-TrOCA0)/a

When O2 is lower, TrOCA is lower, indicating higher remineralization that has to be taken into account to correct $C_T$ data. If one observes the same $C_T$ at year 1 and at year 2, but lower O2 at year 2, then $C_{nat}$ ($C_T$-$C_{ant}$) at year 2 is higher (i.e. $C_{ant}$ is lower).

Line 716: Maybe replace "filtered" by "excluded" or "discarded"
Response: Thank you, this will be corrected: (anomalies in 1998, 2005 and 2020 excluded).

Line 764: missing "uatm"
Response: Thank you, µatm added on line 764.

Line 765: If I understand correctly, a correlation between Chl-a and fCO2 is built into FFNN. Is such a correlation also seen directly in the data? I'm asking because if not, how do you know it is a real feature and not an extrapolation artifact?
Response: The relations presented in figure 8 are constructed from mean summer Chl-a (from SeaWIFS and MODIS) and $fCO_2$ or $C_T$ from the FFNN results. This is a real feature, as in the FFNN model, Chl-a, as well as SST or mixed-layer, is a predictor used to reproduce fCO2 (when fCO2 is not available (i.e. no direct correlation is built in the FFNN). Here the link is presented for the seasonal $fCO_2$ or $C_T$ amplitude. Unfortunately, there is no data each year both in summer and winter for a more direct investigation. Thus, we are not able to evaluate the seasonal amplitude (winter-summer) based on observations, instead using FFNN outputs to explore the link between $fCO_2$ and $C_T$ seasonality and the Chl-a records. Based on the FFNN model, we observed the inter-annual variability of $C_T$ in summer whereas in winter the $C_T$ follows the $C_{ant}$ uptake (figure S12). Therefore we interpreted the change of $C_{nat}$ ($C_T$-$C_{ant}$) seasonality due in part to biological processes

Lines 965-981: It took me quite a while to understand how to read Fig 12, though in the end everything makes good sense. Maybe there is a way to further help readers? I cannot offer a good suggestion either, unfortunately.
Response: Thank you for highlighting this figure; here we aimed at showing on the same plot the properties versus depth or versus time; such a plot was also used in another publication (Metzl et al.

2010, see Figure R4 below) and we thought it was the best way to describe and link the changes in time, in season, and at depth based on observations and FFNN model.

Line 1066: Consider writing "CT(1962)+Cant(t)" in both caption and figure legends, because otherwise the sum "CT+Cant" is quite confusing
Response: Thank you for the suggestion. The caption will be revised:
"The red line is the atmospheric fCO2 and red dashed-lines in each plot are the evolution of properties since 1960 corrected to Cant where fCO2, pH, Ω-ar and Ω-ca were recalculated using CT(1962)+Cant(t)". We also changed the legend in figures.

Line 1096-1097: Couldn't this hypothesis be checked by the FFNN, by calculating the FFNN response to a counterfactually constant SST predictor?
Response: Here we discussed the fCO2 change for the October-November season (same season for the 1962 data). It appeared that fCO2 computed with Cant trend (dashed red line in figure 13) was coherent with observations and FFNN model before 2014 but not after 2015, suggesting that another process explained the more recent fCO2 variability (at least for this season). We suggested that this was due to warming taking place during the recent period. The Reviewer suggests testing the impact of the warming after applying constant SST on FFNN. We have tested this hypothesis by calculating fCO2 with constant SST (at 2°C); this is presented in this review in Figures R2 and R3. When SST is constant, fCO2-2C (orange lines) is close to fCO2 deduced from Cant accumulation after 2016; note also a large anomaly of fCO2-2C in November 2014 when there was a cooling (Figure S9 in the paper) that might be the signal of deeper mixing as suggested from the maximum CT from FFNN (Figure 13 in the paper).

As mentioned in the article, the differences between fCO2 and fCO2 computed from CT+Cant are not very large given the uncertainty in the CO2sys calculation when using AT-CT pairs (+/-13 µatm); we thus cannot interpret the signals beyond what was indicated in the paper. Cruise sampling is marginal for the interpretation of 2-5 years signals, but not for long-term variability (here since 1962) for which our conclusions hold.

[Figure]

Figure R2: Same as for Figure 13 in the paper with fCO2 from FFNN also at constant SST (2°C) in orange.

[Figure]

Figure R3: Same as for Figure R2 for years 2000-2021. In 2017-2021 the warming explains why fCO2 was higher than fCO2 due to Cant uptake only.

Line 1110: "undersaturation"?
Response: Thank you, corrected

Line 1312: "BGC-Argo"
Response: Thank you, corrected

;;;;;;;;;;;;;; Reference in this review:

Lauvset, S. K., Lange, N., Tanhua, T., Bittig, H. C., Olsen, A., Kozyr, A., Álvarez, M., Azetsu-Scott, K., Brown, P. J., Carter, B. R., Cotrim da Cunha, L., Hoppema, M., Humphreys, M. P., Ishii, M., Jeansson, E., Murata, A., Müller, J. D., Perez, F. F., Schirnick, C., Steinfeldt, R., Suzuki, T., Ulfsbo, A., Velo, A., Woosley, R. J., and Key, R.: The annual update GLODAPv2.2023: the global interior ocean biogeochemical data product, Earth Syst. Sci. Data Discuss. [preprint], https://doi.org/10.5194/essd-2023-468, in review, 2024.

Metzl, N., A. Corbière, G. Reverdin, A. Lenton, T. Takahashi, A. Olsen, T. Johannessen, D. Pierrot, R. Wanninkhof , S. R. Ólafsdóttir, J. Olafsson and M. Ramonet, 2010. Recent acceleration of the sea surface fCO2 growth rate in the North Atlantic subpolar gyre (1993-2008) revealed by winter observations, Global Biogeochem. Cycles, 24, GB4004, doi:10.1029/2009GB003658.

[Figure]

**Figure 5.** DIC/TA ratio as a function of depth for the northern and southern region in the NASG (mean of GLODAP data along Suratlant line, open symbols, left axis) and DIC/TA ratio as a function of time for the same regions (mean of Suratlant data, filled symbols, right axis). The DIC/TA ratio are higher in the south (in both data sets), and in recent years this ratio has increased in surface waters.

Figure R4: Figure 5 in Metzl et al 2010, for the plot of properties as a function of depth or time (somehow like Figure 12 in the paper).[DIC/TA ratio as a function of depth for the northern and southern region in the NASG (mean of GLODAP data along Suratlant line, open symbols, left axis) and DIC/TA ratio as a function of time for the same regions (mean of Suratlant data, filled symbols, right axis). The DIC/TA ratio are higher in the south (in both data sets), and in recent years this ratio has increased in surface waters.]

;;;;;;;;;;;;;;;; end reply review 1 ;;;;;;;;;;;;;;;;

---

## Author Comment (AC2)

Replies to reviewer 2 (Her/his Report published 25/1/24)

Article : Metzl, N., Lo Monaco, C., Leseurre, C., Ridame, C., Reverdin, G., Chau, T. T. T., Chevallier, F., and Gehlen, M.: Anthropogenic CO2, air-sea CO2 fluxes and acidification in the Southern Ocean: results from a time-series analysis at station OISO-KERFIX (51°S-68°E), EGUsphere [preprint], https://doi.org/10.5194/egusphere-2023-2537, 2023.

The reviewer's comments and questions are written in black; our replies are in red.

Review of Metzl et al: Anthropogenic CO2, air-sea CO2 fluxes and acidification in the Southern Ocean.

The paper describes a comprehensive data set of in situ observations spanning multiple decades that is backed up by a gap-filled surface ocean pCO2-product in the Indian sector of the Southern Ocean. It provides critical numbers for anthropogenic CO2 uptake and accumulation, acidification rates etc. It also uses an older data set from the 1960s to put the data into perspective and uses the relationship between atmospheric CO2 and anthropogenic carbon to extrapolate the results into the future based on atmospheric CO2 of two emission scenarios. This is a very valuable scientific contribution and I fully support its publication. Thank you for collecting these data and making them available. It is particularly interesting to see the decadal variability in the carbonate system variables and how an increase in biological production can compensate the anthropogenic CO2 uptake in recent years to lead to a relatively stable carbonate system. I have a couple of comments and clarifying questions that I would like the authors to address. The list is lengthy, but many are detailed comments on sentences being not clear.

Response: We warmly thank the reviewer for her/his supportive report

General:
1) The paper is rather long and very detailed. It is certainly a strength of the paper to provide all numbers (convincing evidence) and this is appropriate for a publication in Ocean Sciences. I would nevertheless recommend the authors to see if the text can be somewhat shortened and stream-lined, for example by moving some of the numbers into a table.

Response: Thank you we have reduced the reports of numbers in the text. For a synthesis view we only included the trend values for fCO2 and pH in Table 1 and thought this was useful information when submitted the paper. However, as suggested, we have added a new table (Table 2, below) with the results from our study (with trends for fCO2, pH and CT for each period/season). Table 1 now includes only results from previous studies. The text has been revised accordingly and Table 2 moved to Table 3.
* * *
Table 2: Trends of oceanic $fCO_2$ ($\mu atm.yr^{-1}$), pH ($TS.decade^{-1}$) and $C_T$ ($\mu mol.kg^{-1}.yr^{-1}$) at the OISO-KERFIX location (50°40'S-68°25'E) in the Southern Indian Ocean for different periods based on observations (Obs.) and the FFNN model (FFNN). Standard-deviations are given in brackets.
* * *
| Period | Season | Trend $fCO_2$ $\mu atm.yr^{-1}$ | Trend pH $TS.decade^{-1}$ | Trend $C_T$ $\mu mol.kg^{-1}.yr^{-1}$ | |
|--------|--------|--------|--------|--------|------|
| 1962-2016 | November | 1.31 (0.32) | -0.014 (0.002) | 0.47 (0.01) | Obs. |
| 1991-2021 | Summer | 2.10 (0.22) | -0.022 (0.002) | 0.57 (0.16) | Obs. |
| 1991-2001 | Summer | 0.76 (0.90) | -0.009 (0.010) | 0.05 (0.64) | Obs. |
| 2001-2010 | Summer | 3.23 (1.07) | -0.035 (0.011) | 1.03 (0.77) | Obs. |
| 2010-2020 | Summer | 0.84 (0.77) | -0.008 (0.008) | 0.70 (0.68) | Obs. |
| 1985-2020 | Summer | 1.71 (0.08) | -0.018 (0.001) | 0.68 (0.05) | FFNN |
| 1991-2020 | Summer | 1.85 (0.11) | -0.020 (0.001) | 0.68 (0.07) | FFNN |
| 1991-2001 | Summer | 1.18 (0.26) | -0.013 (0.004) | 0.60 (0.30) | FFNN |
| 2001-2010 | Summer | 2.87 (0.25) | -0.030 (0.003) | 1.08 (0.24) | FFNN |
| 2010-2020 | Summer | 0.98 (0.40) | -0.010 (0.004) | 0.38 (0.26) | FFNN |
| 1985-2020 | Winter | 1.64 (0.05) | -0.017 (0.001) | 0.55 (0.04) | FFNN |
| 1991-2020 | Winter | 1.78 (0.15) | -0.018 (0.001) | 0.56 (0.05) | FFNN |
| 1991-2001 | Winter | 0.98 (0.09) | -0.010 (0.001) | 0.18 (0.14) | FFNN |
| 2001-2010 | Winter | 1.99 (0.10) | -0.021 (0.001) | 1.02 (0.12) | FFNN |
| 2010-2020 | Winter | 2.21 (0.17) | -0.022 (0.002) | 0.69 (0.30) | FFNN |
| 1985-2020 | Annual | 1.57 (0.03) | -0.0165(0.0004) | 0.58 (0.05) | FFNN |
* * *
a. In particular, I would ask the authors to shorten the summary section, which does not need to present all numbers again.

Response: Thank you we agree; we have reduced the summary section

b. Also, I was a bit confused about the sections 3.3.1 "surface pH trend" and "3.3.2 Temporal changes in the water column". The "surface" section 3.3.1 also comments on the depth profiles, which should be covered in the following section.

Response: The reviewer is correct, in section 3.3.1 we also informed on some results in the water column to link results with observed changes at the surface. This is because estimates of Cant in the WW layer are used as representative of the change at the surface during winter. The main section 3.3 was titled "Anthropogenic CO2 drives acidification in surface and the water column". For clarity, we separated sections 3.3.1 and 3.3.2 where the focus of results and discussions mainly deal either with surface (3.3.1) or water column (3.3.2).

2) The figures are not of highest quality, probably plotted in Excel? Maybe consider whether there is a way to use a higher-quality plotting software for future work. I will list specific comments below, on font size, colors, missing legend entries where applicable. Font sizes are also generally too small in the supplementary figures.

Response: Thank you we have revised the figures.

3) Given the availability of a decadal time-series, could you comment on the discussion of the variability of the Southern Ocean carbon sink? E.g. the stagnation of the Southern Ocean carbon sink in the 1990s and/or the reinvigoration in 2015 (Le Quere et al., 2007, doi: 10.1126/science.1136188 Landschützer et al., 2015, doi:10.1126/science.aab2620). You touch on SAM, but I would appreciate a clear statement on decadal variability and its drivers in the summary/concluding section.

Response: The analysis is focused on one location (HNLC in the POOZ) where we have observations to investigate inter-annual to decadal changes in both surface and water column. The SAM was introduced to recall that it might have an effect on the pH trends at local scale (Xue et al, 2018). Therefore, we

looked at the potential link between the SAM and the variability of the CO2 fluxes, CT and pH. As the SAM may also affect upwelling/circulation, we explored the variability of all properties in the water column (which was not investigated by Xue et al, 2018). We found an unclear relation to SAM of our subsurface data. We noted (line 747-760) that in 2001-2010 we did not detect any clear change at depth for ocean properties (except for CT and Cant) that would support a link with the SAM (through enhanced upwelling) and we suggest that change of CT, fCO2 and pH trends are probably caused by surface processes rather by than changes in the water column. However, we cannot eliminate the possibility that SAM influenced changes in biological processes after 2008-2010 (Lines 1282-1283), although at that stage we are not able to quantify this link proposed through modeling studies (Lovenduski and Gruber, 2005, referred in line 616). We did not extend the analysis for the carbon sink to the Southern Ocean basin scale as it has been investigated in other papers (e.g. Hauck et al, 2023 in the frame of RECCAP2). As noted by the reviewer, the paper is "rather long and very detailed". We think it would be appropriate to re-investigate the Southern Ocean carbon sink variability and carbonate systems at large scale using new data-base products such as the one recently developed by Chau et al (2024).

Specific comments:
1) There are quite a few language issues, I will mention some in the technical corrections, but the list is not exhaustive. Also, quite a few times, the sentences weren't super clear. See comments below. In general, please double-check whether it is always crystal-clear whether you talk about numbers/trends from in situ data or from FFNN.
Response: Thank you we have revised the text and languages issues and clarify when we refer to observations or results from FFNN.

2) Line 21: "At the surface during austral winter the oceanic fCO2 increased at a rate close or slightly lower than in the atmosphere. " I'd appreciate if you added what that means for the ocean sink.
Response: This was referring to the fCO2 trend in winter suggesting the ocean sink during this season is not changing much from year to year as the ocean fCO2 somehow tracks the rise of atmospheric CO2 (see figure R1 in this response). We don't think that we need to add more information in the abstract.

[Figure]

Figure R1: Time-series of ΔfCO2 (µatm) (fCO2oce-fCO2atm) and air-sea CO2 flux (mol C/m2/yr) from the LSCE-FFNN model in January (red) and August (blue) at OISO-KERFIX location. During austral winter (August) because the ocean fCO2 tracks the atmospheric CO2 the flux (source) is relatively constant, 1.04 (±0.33) mol/m2/yr in 1995-2020 whereas in summer the sink varies between -2 and +0.4 mol/m2/yr.

3) Line 28 "desperately". I suggest changing to "is expected to increase" or such. How atmospheric CO2 evolves depends on the decisions made by human societies.
Response: Thank you, we agree. Change as suggested

4) Line 44 and elsewhere: why do you use parentheses around uncertainties? They belong to the number, so I would just report them in plain text (no parentheses)
Response: Thank you, in other publication we used parentheses (for ESSD journal) but for Ocean Science, we should not. This has been corrected.

5) Line 45: could update to Friedlingstein et al., 2023, 10.5194/essd-15-5301-2023
Response: Thank you, the most recent GCB paper was published on 5 December 2023, after we submitted our paper. In the introduction we refer to Friedlingstein et al., (2022) and specifically for the decade 2012-2021 that corresponds to the period of our analysis. We prefer to keep the number listed for GCB 2022 (Friedlingstein et al., 2022) and we added the following: "This partitioning has been confirmed for the decade 2013-2022 (Friedlingstein et al., 2023)." Reference also added.

6) Line 54: "ongoing debate" add "on…" (explain the debate a little bit).
Response: Thank you, we added: "but there is an ongoing debate on the size of the carbon sink in this region depending the periods and methods (Long et al., 2021; Sutton et al., 2021; Hauck et al, 2023b; Gray, 2024)." Here we have also added the reference to Hauck et al, 2023b recently published (see also response to comment 8).

7) Lines 55-64: please mention that there is some doubt about the previously reported magnitude of decadal variability in the Southern Ocean carbon sink based on tests with the mapping products (Gloege et al., 2021, 10.1029/2020GB006788, Hauck et al., 2023, 10.1098/rsta.2022.0063)
Response: Thank you for this suggestion. We now refer to Gloege et al (2021) and add the following in this section: "However as for the mean state, there are also uncertainties on both the magnitude and phasing of decadal variability in the SO carbon sink mainly due to insufficient sampling (Gloege et al, 2021; Hauck et al, 2023a,b)."

8) Line 57 here and elsewhere you cite Hauck et al. 2023: "Sparse observations…", doi: 10.1098/rsta.2022.0063 though you probably meant to refer to the RECCAP chapter (Hauck, J.; Gregor, L.; Nissen, C.; Patara, L.; Hague, M.; Mongwe, P.; Bushinsky, S.; Doney, S. C.; Gruber, N.; Le Quéré, C.; Manizza, M.; Mazloff, M.; Monteiro, P. M. S.; Terhaar, J. The Southern Ocean Carbon Cycle 1985–2018: Mean, Seasonal Cycle, Trends, and Storage. *Global Biogeochemical Cycles* **2023**, *37* (11), e2023GB007848. hUps://doi.org/10.1029/2023GB007848.)
Response: Thank you, reference to Hauck et al 2023a,b added, they both informed on the decadal change and uncertainties.

9) Line 72: upwelling - enhanced upwelling
Response: corrected

10) ~Line 74: I would personally add a note on the suggested secondary saturation horizon that is expected as the anthropogenic carbon uptake is strongest at the surface (Hauck et al., 2010; Negrete-Garcia et al., 2019). You seem not to confirm this with your results, so might be worth introducing this here in the first place.
Response: In the introduction, we refer to suggestions that the "large scale" under-saturation state in the SO might be reached as soon as 2030-2050. In this introduction we did not refer to Hauck et al (2010), a study focused in the Weddell Sea from 3 cruises in 1992, 1996 and 2008 which suggested that the surface under-saturation in this region will be reached after the 21st century. Later, in the results section (lines 943-945) we mentioned the rapid under-saturation horizon as suggested in other analyses (Hauri et al., 2015; Negrete-Garcia et al, 2019) but not observed in our data at KERFIX location. No change.

11) Line 90: "modest source of CO2": Combined with the next sentence it reads as if this is a model flaw. A large sink certainly is, but a modest source is possible I would say. We have very few preindustrial observations ;) and so don't know the number well.
Response: Yes, here we referred to the uncertainties in the models. No change.

12) Line 91-98: I would tone down the comment on the models a bit, because, yes there are systematic flaws, but also, often they fall into the observational range and/or we don't really have the necessary process understanding that would be needed to cure those uncertainties/biases.
Response: We agree that models can fall into the observational range especially when compared to annual fluxes but we wanted to point out that this is often for wrong reasons, i.e. the seasonal flux is not correct due to bad representation of $C_T$ seasonal cycle as indicated in this section. No change.

13) Line 175: can you comment on whether the Leseurre et al paper is based on exactly the same paper?
Response: The Leseurre et al paper investigated the surface trend of fCO2 and pH in 1998-2019 for summer in the south Indian (i.e. using surface data in summer). Here we extended the analysis back to 1985 and to 2021 (also using data from 1962) in surface and the water column and for all seasons using results of the FFNN model. We think this is clearly explained. No change.

14) Line 190: maybe add Hauck & Völker, 2015, 10.1002/2015GL063070
Response: Thank you for the suggestion. Reference added.

15) Figure 1: please increase fontsize of all text on axes and within figure. Yellow text "Kerguelen" is a bit hard to read
Response: Thank you, figure revised (size text and Kerguelen in white)

16) Line 289: please give a quick summary of accuracy of 1990-1995 data
Response: The accuracy of the data from 1990-1995 is indicated in section "2.2.3 Data quality-control and data consistency." We recalled that the AT-CT data were corrected and available in GLODAP. Based on comparison with other cruises, the AT-CT data have been revised in the GLODAPv2.2019 version as indicated lines 337-341. We compared the data in deep layers with the most recent data at that station, and found coherent concentrations (Supp Mat., Table S2, Figure S1). In conclusion, we believe the AT-CT data in 1990-1995, when available, have an accuracy of 4 umol/kg (the value indicated in GLODAP).

17) Please comment on accuracy of 1960s data
Response: The $fCO_2$ data from 1962 were obtained from the SOCAT data-set. The cruise in 1962 was flagged "D" during the quality control process by SOCAT (as indicated in Table S1b) and has thus an accuracy better than 5 µatm. As suggested by the reviewer, we have added this information on line 260: "All surface temperature, salinity and fCO2 data were extracted from the SOCAT data-product version v2022 (Surface Ocean CO2 Atlas, Bakker et al., 2016, 2022) and have an fCO2 accuracy between 2 to 5 µatm".

18) Line 298: CRMs were used for all data since 1998 and for none before that?
Response: CRMs were also used in 1993 during KERFIX cruises (Louanchi et al, 2001) but as indicated above the AT-CT data in 1993 were corrected by 35 to 50 µmol/kg. No change.

19) Line 322: can you please add the section number where this is shown.
Response: Good suggestion, section 3.1 added in line 322.

20) Section 2.2.3: please comment on 1960s data as well.
Response: In section 2.2.3 we discussed the quality control of AT-CT data. In 1962 there is only data for surface fCO2. No change.

21) Line 359: suggest to delete the sentence on pH here, as this is the topic of the next paragraph and alkalinity has not yet been introduced.

Response: This section 2.2.4 introduces the LSCE-FFNN model. We keep the information that pH is calculated using FFNN results. No change.

22) Line 419 "although": add that there also wasn't much winter data for training?
Response: Thank you, this is added as follows: "although the FFNN model was not constrained by in-situ fCO2 before 1991, few data in austral winter since 1991, and no Chl-a satellite data available before 1998."

23) Line 420-423: I'm confused about this sentence. Why is it here and what is the message? This is the methods section, not results.
Response: This section is dedicated to comparisons with different datasets and the FFNN model. We explained that the model was not constrained by data before 1991, but we think a presentation of the comparison with 1985 data is useful (Figure S4). We agree with the reviewer concerning the last sentence in this section. It has been deleted and moved in section 3.1.

24) Table 1 and Figure 2 provides averages over certain time-periods, but these are not defined/explained until the end of section 3.1. Please explain this earlier (methods, even if referring to results). I am also puzzled over the chosen *five* seasons, some of them being single months. Why not stick to DJF, MAM, JJA, OND? This needs justification. Also for the time-periods considered, it would be useful to also report numbers for "standard decades" 1990-1999, 2000-2009 etc even if in supplement. The chose time periods also use the "edge" years twice, e.g. 2001-2010 and 2010-2020 both have 2010 included. Is that on purpose?
Response: The choice of periods for averaging the observations or to estimate the trends depends on the original data available that are first presented in section 3.1. We don't think this should be explained in the method section. No change.

The reviewer is correct about the "five" seasons. Given the data available at different periods (years and months) and the seasonal cycle of temperature, fCO2, or CT, we distinguished between five periods (called seasons in the legend). For clarity we change "five seasons" to "five periods of the year" in the caption.

The trends: we have reported results for different seasons in 1985-2020, 1991-2020, 1991-2001, 2001-2010, 2010-2020, 1991-2001, 2001-2010, 2010-2020, and 1985-2020 (new Table 2). We are not sure why the reviewer asks for "standard decades", i.e. 1990-1999, 2000-2009. We think the paper includes sufficient information on the trends (based on observations or FFNN model) and a selection of 3 periods to discuss the drivers.

Concerning the "edge" in the selected periods, the reviewer is correct: given the data available in surface and subsurface (for Cant estimates) we included 2010 for both 2001-2010 and 2010-2020 trends. We used the start and the end for each period.

As suggested in section 3.1 we have added a sentence to better explain the choice of the periods selected as follows: "The model estimates a decrease of the annual CO2 source until the end of the 1990's followed by an increase of the source over the following decade (Figure 3). Around the year 2010, the annual CO2 flux was around +0.5 molC.m-2.yr-1 and then decreased over the last decade to change into an annual CO2 sink that increased to reach -0.5 molC.m-2.yr-1 in 2020. For this reason and given the data available since 1991, we evaluated the summer and winter trends in fCO2, CT and pH from the FFNN model over 3 periods 1991-2001, 2001-2010, 2010-2020 and compared the summer trends with those deduced from observations (Table 2)." Note this is new table 2 (see above) and in Table 2, we have added the trend for 1991-2020 and 1985-2020 in winter for FFNN (not listed in the submitted MS)

25) Line 430: "sampling locations were mainly reoccupied in austral summer": unclear, where do the data for different seasons then come from? Do you simply mean: "...different seasons, though most of them stem from summer"?

Response: We wanted to point out that most of the cruises were conducted in summer (both for surface and station data). As suggested the sentence is revised: "fCO2 measurements are available for different seasons since 1991, though most of them stem from austral summer (January-February)."
We specify austral summer (January-February) for those not familiar with the SO.

26) Figure 2 and 4: please increase font size of all text (axes labels, legend). The inserted map is far too small. Colors of the symbols in the legend are not readable. Suggest changing the purple to grey or such, red symbols on top of purple line are hard to read. Still puzzled about the chosen seasons (that don't seem to be interpreted much) and decadal periods.
Response: Thank you these figures have been revised. The grey line is a good choice. The map has been enlarged. Figure S13 in the Supplementary Material has been also revised.

27) Line 487: looks to me the positive SAM started rather in 2008 than in 2010. There were also positive phases earlier.
Response: This is correct, the SAM is positive in 2008 but negative in 2009; here we pointed out the periods when SAM was positive (higher than 0.5) during several years. This does not change the interpretation. There was also high SAM index in 1982 or 1993 but we did not discuss this for the trends analysis. Please note that there was an error in the legend. For the wind we also presented the 24-month running mean, similarly to what is done for the SAM. This has been corrected in the legend. For clarity we also slightly changed the figure, by adding the "0.5" limit for the SAM.

28) Line 506: "part of": how much?
Response: The $A_T$ concentrations in October 2016 were higher by 6.2 µmol/kg compared to 1993. This reduced the $fCO_2$ by 16.3 µatm. The $fCO_2$ calculated in October 1993 was 342.8 µatm and in October 2016 $fCO_2$ calculated was 396.4 µatm, very close to that measured (396.7 µatm). The difference of +53.5 µatm would result in a trend of +2.3 µatm/yr over 23 years, close to that derived from $fCO_2$ measurements in 1995 and 2016 (2.1 µatm/yr). If $A_T$ was constant, the $fCO_2$ in 2016 would be higher, 412.6 µatm, and the difference of +69.8 µatm would result in a trend of 3.0 µatm/yr, i.e. +0.7 uatm/yr faster. Accordingly the sentence (line 506) would have been revised: "Part of the difference (+0.7 µatm.yr$^{-1}$) may be explained by AT that was slightly higher (+6 µmol.kg-1) in October 2016". However, to shorten the paper (one of your general comment), lines 497-507 have been deleted in the revision.

29) Figure 3: increase font sizes.
Response: Thank you, figure revised.

30) Figure 5: increase all font sizes. Change decimal comma "," to "." Orange dashed line missing in legend.
Response: Font sizes changed. The orange dashed added in legend

31) Line 590: "stability" this needs explanation before figure is shown, or delete in the caption.
Response: We agree, this is deleted in the caption.

32) Line 600: "albeit…" does this refer to October? Unclear.
Response: Yes, this is to indicate that in 2010-2021, there were only two cruises in October (in 2011 and 2016, Table S1b). Sentence revised: "This rate is also lower compared to the change observed in October (+2.9 µatm.yr-1) albeit being only based on 2 cruises in October 2011 and 2016 (Figure 2a)."

33) Line 614-615: how does this increasing chl fit with no trend observed in nutrients? Everything used up immediately? Hauck et al., 2013 also found in a modelling study that increasing NPP can counteract the outgassing during a positive SAM phase. Would be nice to at some point in the discussion comment on the system response to SAM (summarize your findings).
Response: We have no clear understanding of the link of the SAM with nutrients (see also our response to your comment 77). As opposed to $fCO_2$ we have no full coherent time-series for nutrients (observations or reconstruction). This is an interesting topic that might deserve another analysis. Based on our data we can only conclude that during specific years (e.g. 1998) we measured very low silicates (<

2 µmol/kg down to 100m, Figure S6) (discussed lines 555-562), and that might be a response to the SAM or the ENSO or both (e.g. the strong El Nino in 1997-1998, Jabaud-Jan et al, 2004). As far as we know no model (OBGM or 1D) is able of reproducing silicate concentrations as low as 2 µmol/kg in the SO in the surface waters or at depth as those we observed in 1998 and 2014. This suggests that biological parameterization (iron limitation, light, etc…) in models should be revisited or tested. Concerning the SAM and the $C_T$ changes, we referred to Hauck et al (2013) on line 751, and this is a good idea to also refer to this publication on line 615. We thus have added: "Modeling studies also suggest that summertime biological activity could play an important role to explain the variability of the $CO_2$ sink in the SO in the response to the SAM (Hauck et al, 2013)."

34) Line 627: "higher": higher than what? It is still well below atm CO2.
Response: Thanks for the question. $fCO_2$ in 2020 was higher than $fCO_2$ in 2019 (although SST was lower). Revised as follows: "In 2020, although the temperature was also lower than in 2019, the oceanic fCO2 was higher (392 µatm)"

35) Line 637: this information is needed much earlier. I would also welcome a more detailed justification of those periods, it seems a bit arbitrary here.
Response: In previous sections we described the $fCO_2$, $C_T$ observations and FFNN results in surface waters along with air-sea $CO_2$ fluxes, SST, Chl-a variability in summer. With the data at hand and for analyzing the trends and their drivers, this description leads to separating the time-series into 3 periods and we thought it was appropriate to list these periods at the end of this section. As noted above (response to comment 24), we have added a sentence to explain the selection of periods at start of section 3.1 and Lines 636-638 deleted.

36) Line 649: add depth range (100-300 m?)
Response: The Winter water is mainly found in the depth range 150-250m. Information added on line 649.

37) Figure 6: Increase font sizes of axes and color bar labels. I know Cant detection methods can result in negative values, but it would be good to comment on them (artefacts or real?) What is the lowest value that you find?
Response: Figure 6 has been revised. The negative Cant values were not so numerous in our time-series (Figure R2 below). The lowest Cant values was -5.6 µmol/kg at 1603m (in 2015). This is in the range of the uncertainty for the TrOCA method (6.5 µmol/kg, indicated line 679). The highest $C_{ant}$ value was 79.3 µmol/kg at 118m (in 2021) but the results are probably less reliable in summer in the top layers (due to biological activity).

[Figure]

Figure R2: $C_{ant}$ profiles from all cruises showing 5 negative values estimated in deep waters. Color code is year.

38) Line 674: "top layers (0-400m, Figure 6b): Figure 6b only starts at 200 m…
Response: Thank you, this is corrected.

39) Line 679-681: suggest ordering from surface to depth, so change order of these two sentences.
Response: Thank you for this suggestion. Order changed.

40) Line 693: "natural variability". I'm curious whether more details on processes at work can be given?
Response: At that stage, we have no clear explanation of the low $C_{ant}$ occasionally observed (in 1998, 2005 or 2020). For these periods we only identified some anomalies in O2, but not for other properties (T, S, AT). We also looked at measured currents (VM-ADCP) when available (https://doi.org/10.17882/88407) but there were no specific anomalies and we thus did not introduce the information on current data in the paper. As the anomalies are not linked to an anthropogenic signal, we referred to it as "natural variability" but cannot further specify the processes (biology, circulation, mixing, other ?).

41) Line 695: explain links between CT, O2, temp and Cant here.
Response: Reviewer 1 has a similar comment.
To calculate $C_{ant}$ concentrations in the water column, one separates the contribution of $C_T$ change due to organic matter remineralization. This is achieved using O2 data with an adapted Redfield ratio; following the concept of a quasi-conservative tracer such as NO (Broecker, 1974), TrOCA is a tracer based on a combination of CT, AT and O2:
$$TrOCA = O2 + a (C_T - A_T/2) \text{ and } C_{ant} = (TrOCA - TrOCA0)/a$$

When O2 is lower, TrOCA is lower, indicating higher remineralization that has to be taken into account to correct $C_T$ data. If one observes the same $C_T$ at year 1 and at year 2, but lower O2 at year 2, then $C_{nat}$ ($C_T - C_{ant}$) at year 2 is higher (i.e. $C_{ant}$ is lower).

Temperature is used to calculate TrOCA0, but has no direct effect on Cant as temperature and TrCOA0 did not vary much in these time series. We have revised the sentence (AT added and temperature deleted): "In 2005 anomalies of CT, AT and O2 concur to explain that we calculated lower Cant (43.9 µmol.kg-1).

42) Figure 7: moderate increase in font sizes
Response: Figure 7 revised font sizes

43) Line 729: I'm curious why none of these data was used by Gruber et al? Is it not in Glodap?
Response: This is because to apply their method (eMLR*) Gruber et al used phosphate data. The OISO cruises do not include phosphate data for most years and Gruber et al therefore did not select them.

44) Line 744-746: "CT trend faster in summer <-> negative CT anomaly" doesn't make sense to me, more CT or less?
Response: We pointed out that because of the low CT (negative anomaly) associated to the high production in 1998, the $C_T$ trend in 1991-2001 was larger in summer compared to the $C_T$ trend in winter and the $C_{ant}$ trend in subsurface. However, filtering the 1997-1998 summer anomalies did not change this view, i.e. the $C_T$ trend in 1991-2001 was faster than the trend in Cant. Therefore we have changed the description as follows: "The FFNN model showed that the CT trend in summer was faster than the trend in Cant (Figure 4b), suggesting that natural processes would have increased CT. This could be explained by an increase in vertical mixing due to the increase in wind speed (Figure 3). On the contrary, the winter CT trend was lower than the Cant trend estimated in subsurface waters (Figure 4b)."

45) Line 749: "twice the rate … that could be explained": how much can be explained and how was that estimated?
Response: The $C_{ant}$ trend in the WW of 0.54 µmol/kg/yr for this decade (2001-2010) was also estimated from data in the WW layer and compared with surface $C_T$ trend of around 1 µmol/kg/yr (in summer

from Observations and FFNN and in winter from FFNN). This is about "twice" the $C_{ant}$ trend for this period. As this is observed for both seasons and different from other periods, we first suggested this might be explained by enhanced upwelling (possibly linked to a positive SAM index ?), However, as there was no specific signal in other properties, we suspect this was not linked to upwelling as described later in this section.… and thus we suggested that processes occurring at the surface are the best candidate to explain the rapid $C_T$ trend in 2001-2010 (see also next response).

To clarify the sentence has been revised as follows: "They were also twice the Cant rate in the WW, which could be explained by enhanced upwelling of CT-rich deep waters during this period after the SAM reached a high positive index (Figure 3; Lenton and Matear, 2007; Le Quéré et al., 2007; Hauck et al., 2013)."

46) Line 753: "processes at the surface": which ones?
Response: This is explained later (lines 761-768), the biological activity is the best candidate. The sentence has been revised as follows: "The rapid CT (and fCO2) trend for this decade is probably due to processes occurring at the surface (e.g. biological activity, as discussed later) rather than changes in the water column (vertical mixing or upwelling)."

47) Line 756/57: I'm a bit lost which information comes from in situ obs and which from FFNN.
Response: Thank you. In the revision we now specify that for winter this is from FFNN
"In winter the CT trend (from FFNN) is close to Cant indicative of the anthropogenic accumulation."

48) Line 759: "the temporal change" to absence of temporal change
Response: Thank you, corrected "low temporal change"

49) Figure 8: subscripts for CO2 in the legend would be nice.
Response: Figure revised also for text size.

50) Line 811ff: I am not sure how useful it is to list values from different sources which are all from different periods. This is acknowledged in the next sentence, but maybe this would a good place to shorten and refer to the numbers in the table without repeating them in the text.
Response: We think it is useful to list values when describing the results although some are found in Table 1. Table 1 was dedicated to results based on observations and for results derived from reconstructions (Ma et al, 2023) we prefer to list the value in the text when appropriate for the discussion (now in parenthesis).

51) Figure 9: see comments on Figure 2. I guess TS stands for total scale. Please spell out in the caption.
Response: Figure 9 revised like figure 2.

52) Line 863: the first number is from in situ obs? It would help me and maybe also other readers to specify.
Response: Yes the first number in the WW is from the station data.

53) Line 894: "no trend is observed for pH-PI": then why do you give a trend number in the figure?
Response: The number in the figure for pH-PI is reported to compare it with the trend for pH.

54) Figure 11: increase font sizes. It would be easier to read if a box or background shading would be used to indicate the data for depth < 500 m (instead of the arrow). Otherwise, the arrow should be labelled directly. Also mark and label the data for depth > 500 m.
Response: Thank you, figure revised.

55) Line 935: "no change of Cant". Well, the figure shows between 0 and 30 μmol/kg Cant, this is not 0.
Response: Here when we refer to "no change", this means no change of Cant from 1985 to 2021, i.e. below 500m Cant is not zero but there is no significant increase or decrease during this period.

56) Line 939: "any appreciable": how defined?

Response: The changes below 500m are in the range of uncertainty

57) Line 940: saturation state → saturation horizon!!
Response: Thank you, corrected.

58) Line 947: saturation → saturation horizon
Response: Thank you, corrected.

59) Line 960ff: the percentage numbers, do they really refer to pH or to H+. H+ would make more sense to me given that pH is a logarithmic scale.
Response: Here we refer to pH. If one uses H+ values we got the same results: in the WW in 1985 H+ was 8.8 nmol/kg and in 2021 H+ was 9.9 nmol/kg, i.e. an increase of 1.1 nmol/kg in 36 years. Compared to the H-PI (of 6.4 nmol/kg) this represents an increase of 1.1/3.5 = 31% (about 30% change, same as for pH). The sentence has been revised as follows: "At 200m, based on AT and CT data, we estimated a decrease in pH from 1985 to 2021 by -0.059 (Figure 10b), corresponding to an increase by +1.1 nmol.kg-1 in [H+] (Figure S13), and a decrease by -0.16 in Ωar (Figure S15). Over 36 years, this represents about 30% of the total change since the pre-industrial era for pH (-0.184), [H+] (+3.5 nmol.kg-1) and Ωar (-0.6)."

60) Line 965: carbonate properties → carbonate system properties
Response: Thank you, corrected.

61) Line 969: seasonality → seasonal amplitude
Response: Thank you, corrected.

62) Line 972: I do not understand what this sentence is meant to say.
Response: Here we note that the FFNN results in surface in winter are different from those in summer and close to those observed during summer cruises in the Winter Water.

63) Line 977-981: I am confused about this part, I guess you mean to say that WW layer data can be compared to surface data, but please simplify the sentences.
Response: Yes, we note that the FFNN surface properties in winter are close to those observed in-situ in the WW layer. In other word: data in the WW layer in summer can be compared to surface data in winter. As an example, Figure 12 shows that this is the case for 1985 and 2021. This was also shown in figure S11. Lines 977-981 should also help readers to read figure 12 and for clarity we have modified the sentence: "In winter, at the surface, CT was higher and pH, [CO32-], Ωar were lower (from the FFNN model, blue line in Figure 12). The winter surface values in 1985 and 2020/2021 are in good agreement with observations at depth in the winter water (150-200m). As an example, in 1985 surface CT in winter was 2145.5 µmol.kg-1, which corresponds to the concentration measured at 150m during summer (purple line in Figure 12)."

64) Figure 12: increase font sizes. I found it very confusing to have the FFNN data with time on the y-axis plotted over the profiles of in situ data. After a while I understood, but it would be much appreciated if this could be simplified. An alternative could be to make a second panel for the FFNN data where color of dots is used to mark the years. This could be a smaller panel with only CT on the x-axis, no y-axis and would avoid the impression of a depth profile. Caption refers to "Jan 2020 obs", but only Jan 2021 obs are in legend.
Response: Reviewer 1 made a similar same comment. We would like to keep this view with both depth (for data) and time axis (for FFNN). Figure 12 have been revised (font sizes). We also revised captions (thank you). In the caption both Jan-2020 (black) and Jan-2021 (red) were indicated.

65) Line 1039: does this calculation use salinity from 1962 or a climatology?
Response: Here we used Salinity from climatology. As indicated in Table S1b, there was no salinity data in 1962 and SOCAT used WOA for fCO2 calculation based on original data. As suggested, we have recalled this on line 1039: "The CT concentration, pH, Ωar and Ωca for 1962 were calculated using fCO2

data and AT (from the AT/S relationship Eq. 1) with salinity from World Ocean Atlas (Antonov et al, 2006)."

66) Figure 13: increase all font sizes.
Response: Figure 13 revised and also from reviewer 1 suggestions.

67) Line 1071: 2014: I can't see SST in 2014 sticking out in Figure 13a. (2012?)
Response: In Figure 13a, SST in November 2014 was below 2°C (purple line).

68) Line 1096: did you calculate the effect of the Delta T on pCO2?
Response: Here we discussed the fCO2 change for the October-November season (same season for the 1962 data). It appeared that fCO2 computed with Cant trend (dashed red line in figure 13) was coherent with observations and FFNN model before 2014 but not after 2015, suggesting that another process explained the more recent fCO2 variability (at least for this season). We suggested that this was due to warming taking place during the recent period. The Reviewer suggests testing the impact of the warming after applying constant SST on FFNN. We have tested this hypothesis by calculating fCO2 with constant SST (at 2°C); this is presented in this review in Figures R3 and R4. When SST is constant, fCO2-2C (orange lines) is close to fCO2 deduced from Cant accumulation after 2016; note also a large anomaly of fCO2-2C in November 2014 when there was a cooling (Figure S9 in the paper) that might be the signal of deeper mixing as suggested from the maximum CT from FFNN (Figure 13 in the paper).

[Figure]

Figure R3: Same as for Figure 13 in the paper with fCO2 from FFNN also at constant SST (2°C) in orange.

[Figure]

Figure R4: Same as for Figure R2 for years 2000-2021. In 2017-2021 the warming explains why fCO2 was higher than fCO2 due to Cant uptake only.

As mentioned in the article, the differences between fCO2 and fCO2 computed from CT+Cant are not very large given the uncertainty in the CO2sys calculation when using AT-CT pairs (+/-13 µatm); we thus cannot interpret the signals beyond what was indicated in the paper. Cruise sampling is marginal for the

interpretation of 2-5 years signals, but not for long-term variability (here since 1962) for which our conclusions hold.

69) Line 1110: "extrapolation of trends": do I understand correctly that this is using the trend/year and multiplying with number of years? i.e. independent of atm $CO_2$ assumptions? Please specify.
Response: Yes, for this section we simply extrapolate in the future the trends deduced from 1985-2020 FFNN results (Figure S17) which is independent of the atmospheric $CO_2$ assumptions evaluated in the next section.

70) Line 1140/41: the aragonite saturation → the surface aragonite undersaturation
Response: Thank you, corrected.

71) Line 1145: "correction": this is not a correction, but a sensitivity test. Maybe simply say "warming"? Also note that this offline approximation of temperature will overestimate the temperature effect as it neglects circulation and mixing (warming will be limited by how much not-yet-warmed water is brought into the surface mixed layer)
Response: Thank you, changed "correction" by "warming". We agree that for future scenarios, we are not considering a change in circulation or mixing. For such analysis, one would need to use an ESM model. This is why we mentioned that our analysis excludes changes in ocean circulation (lines 1125-1126 and 1305).

72) Table 2: I am a bit lost here. First: why is SSP2-4.5 (the more realistic emission scenario) only shown for winter, and not also for summer (and the test cases)? Another idea might be to list the test cases in another table so that the design of these can be grasped quicker. Short for SSP2-4.5 is usually ssp245 and for SSP5-8.5 it is ssp585 (ssp85 → ssp585). I also overlooked the ssp45 in the table entirely for a while. Could you add a white space between the lines with the high and low emission scenario?
Response: We have selected two scenarios to get a range of change of carbonate system properties in the future and compare them with other studies based on observations or ESM using the same scenarios (Midorikawa et al., 2012; Jiang et al; 2023). We don't really know if the SSP2-4.5 is a more realistic emission scenario (this will depend on decisions at COP29, COP30, COP31, etc…). To get the most "dramatic" potential change in the future we used the high emission scenario for sensitivity analysis and to compare summer versus winter (as noticed by Sasse et al, 2015). This shows that future change is faster in winter and this is why we only present the results for winter with the low emission scenario. Adding results for summer for SSP245 would add more curves in figures 14 or 15 and would not change the conclusions. Table 2 (now Table 3) has been corrected (acronyms and white space as suggested).

73) Figure 14 and 15: increase font sizes, use subscripts in legend, use complete name for scenarios.
Response: Thank you, Figures 14 and 15 revised (text, acronyms).

74) Line 1260: the → anthropogenic
Response: Thank you, corrected

75) Line 1264: at minimum → at its minimum.
Response: Thank you, corrected

76) Line 1269: detectable increase: of what? Concentration or trend?
Response: Here we refer to the increase of the trend (for concentrations this is expected and listed on line 1269). This calls for maintaining the observations to find if this signal is still increasing in the next decade or so.

77) Line 1283: interesting, and it would be nice to comment on the complete effect of positive SAM index on the $CO_2$ system. Upwelling and outgassing limited to further south? At this location mostly nutrient effects?

Response: Our results at one location in the POOZ/HNLC should be extended at larger scale to investigate (or revisit) the effect of positive SAM on upwelling and its impact on outgassing and carbonates system. This would deserve another study (extending the data-set and probably using an OBGM for sensitivity tests).

Concerning the nutrients, we did not detect a signal that could directly be related to the SAM (Figure R5), although data are more in favor of an anti-correlation.

[Figure]

Figure R5: Silicates concentrations observed in surface at station OISO-KERIFX in summer (red circles) and the SAM index (grey, 12-month running mean). When SAM is positive in 2011, 2016, 2019 and 2021 silicates were low. When SAM is negative in 2010, 2017 or 2020 silicates were higher. However, low silicates were also observed in 1998 and 2014 when SAM was neutral or slightly positive. In 2002 and 2004, silicates were also high when SAM was positive. Thus, no significant correlation can be inferred from this time-series (which is maybe too short for such investigation).

78) Line 1290: any change of giving some numbers for source and sink from Rödenbeck et al?
Response: Here we invited the reader to read the paper by Rödenbeck et al and specifically their figure 6 for the Indian sector. The numbers are a CO2 source around 0.10 PgC.yr-1 in 1960-1990 and a sink around -0.05 PgC.yr-1 in 2020. This is added in the text.

79) Line 1303: "desperately" see comment 3 above.
Response: Thank you, we agree. Changed as suggested ("is expected to increase")

ICI 27/2

80) Line 1313: "coupling of .. not well represented…" : I think this is a bit too general. The coupling between CT, AT, fCO2 and pH is actually well represented with carbonate chemistry routines. It is more some of physical drivers and most of the biological drivers that cannot be represented with sufficient detail/process understanding.
Response: Thank you, sentence revised as follows: "offer useful data for the evaluation of biogeochemical and Earth system models, especially for the physical and biological drivers of the carbonate system not well represented in current models at seasonal to decadal scales in the Southern Ocean (e.g. Hauck et al., 2023a; Rodgers et al., 2023; Joos et al., 2023)."

Technical corrections (not complete):
Line 19: In subsurface → In the subsurface
Line 41: grammar, "taking up a large part … since decades"
Line 49: grammar
Line 65: is → in
Line 87: reach 0.7 → reach up to 0.7
Line 89: ESM → ESMs

Line 123: are in bracket → are given in brackets
Line 309: be not → not be
Line 321: latitude → latitudes
Line 343: excepted → except
Line 565: would occurred → would have occurred
Line 742: "count" → contribute to XX
Line 899: grammar
Line 990: the all → all the
Line 1066: corrected to → corrected for
Line 1158/59: grammar
Line 1312: BG → BGC

Response: Thank you for your careful reading. All lines corrected.

;;;;; References in this response not cited in the submitted MS and some added in the revision

Chau, T.-T.-T., Gehlen, M., Metzl, N., and Chevallier, F.: CMEMS-LSCE: a global, 0.25°, monthly reconstruction of the surface ocean carbonate system, Earth Syst. Sci. Data, 16, 121–160, https://doi.org/10.5194/essd-16-121-2024, 2024.

Friedlingstein, P., O'Sullivan, M., Jones, M. W., Andrew, R. M., Bakker, D. C. E., Hauck, J., Landschützer, P., Le Quéré, C., Luijkx, I. T., Peters, G. P., Peters, W., Pongratz, J., Schwingshackl, C., Sitch, S., Canadell, J. G., Ciais, P., Jackson, R. B., Alin, S. R., Anthoni, P., Barbero, L., Bates, N. R., Becker, M., Bellouin, N., Decharme, B., Bopp, L., Brasika, I. B. M., Cadule, P., Chamberlain, M. A., Chandra, N., Chau, T.-T.-T., Chevallier, F., Chini, L. P., Cronin, M., Dou, X., Enyo, K., Evans, W., Falk, S., Feely, R. A., Feng, L., Ford, D. J., Gasser, T., Ghattas, J., Gkritzalis, T., Grassi, G., Gregor, L., Gruber, N., Gürses, Ö., Harris, I., Hefner, M., Heinke, J., Houghton, R. A., Hurtt, G. C., Iida, Y., Ilyina, T., Jacobson, A. R., Jain, A., Jarníková, T., Jersild, A., Jiang, F., Jin, Z., Joos, F., Kato, E., Keeling, R. F., Kennedy, D., Klein Goldewijk, K., Knauer, J., Korsbakken, J. I., Körtzinger, A., Lan, X., Lefèvre, N., Li, H., Liu, J., Liu, Z., Ma, L., Marland, G., Mayot, N., McGuire, P. C., McKinley, G. A., Meyer, G., Morgan, E. J., Munro, D. R., Nakaoka, S.-I., Niwa, Y., O'Brien, K. M., Olsen, A., Omar, A. M., Ono, T., Paulsen, M., Pierrot, D., Pocock, K., Poulter, B., Powis, C. M., Rehder, G., Resplandy, L., Robertson, E., Rödenbeck, C., Rosan, T. M., Schwinger, J., Séférian, R., Smallman, T. L., Smith, S. M., Sospedra-Alfonso, R., Sun, Q., Sutton, A. J., Sweeney, C., Takao, S., Tans, P. P., Tian, H., Tilbrook, B., Tsujino, H., Tubiello, F., van der Werf, G. R., van Ooijen, E., Wanninkhof, R., Watanabe, M., Wimart-Rousseau, C., Yang, D., Yang, X., Yuan, W., Yue, X., Zaehle, S., Zeng, J., and Zheng, B.: Global Carbon Budget 2023, Earth Syst. Sci. Data, 15, 5301–5369, https://doi.org/10.5194/essd-15-5301-2023, 2023.

Gloege, L., McKinley, G. A., Landschützer, P., Fay, A. R., Frölicher, T. L., Fyfe, J. C., et al: Quantifying errors in observationally based estimates of ocean carbon sink variability. Global Biogeochemical Cycles, 35, e2020GB006788. https://doi.org/10.1029/2020GB006788, 2021.

Hauck, J. and Völker, C.: Rising atmospheric CO2 leads to large impact of biology on Southern Ocean CO2 uptake via changes of the Revelle factor, Geophys. Res. Lett., 42, 1459–1464, doi:10.1002/2015GL063070, 2015

Hauck, J., Gregor, L., Nissen, C., Patara, L., Hague, M., Mongwe, P., et al.: The Southern Ocean carbon cycle 1985–2018: Mean, seasonal cycle, trends, and storage. Global Biogeochemical Cycles, 37, e2023GB007848. https://doi.org/10.1029/2023GB007848, 2023b.

;;;;;;;;;;;;; end response review 2